# SOCIA: Joint Structure–Parameter Co-Optimization for Automated Simulator Construction

## Abstract

Building credible simulators from data is difficult because structure design, parameter calibration, and out-of-distribution (OOD) robustness are tightly coupled. We introduce **SOCIA** (**S**imulation **O**rchestration for **C**omputational **I**ntelligence with **A**gents), a framework that treats simulator construction as **joint structure–parameter co-optimization**: it elicits mechanism-rich blueprints, exposes explicit tunable parameters, and instantiates a calibration schema, producing an executable simulator with built-in calibration hooks. SOCIA couples **Bayesian Optimization** for sample-efficient point calibration with **Simulation-Based Inference** for uncertainty-aware fitting; diagnostics trigger targeted structural edits in an outer refinement loop to co-optimize design and parameters under tight budgets. Across three diverse tasks, SOCIA consistently outperforms strong baselines, excelling on both in-distribution (ID) fitting and OOD shift. Ablations that weaken structure, calibration design, or tuning yield near-monotone degradations, underscoring the necessity of unified structure–parameter optimization. SOCIA's code and data are available here: `https://anonymous.4open.science/r/SOCIA-58B0`.

## 1 Introduction

Simulators—computational models that instantiate mechanisms of interacting agents and environments—are central to modern science and policy, as they attempt to gain insight into the unknown, enable explanation beyond prediction, support the exploration of "what-ifs," and provide *ex ante* policy stress-testing in systems where experimentation is costly or unethical (Bonabeau, 2002; Epstein, 2008; Peldon et al., 2024). Their distinctive value lies in generating both in-distribution and credible out-of-distribution (OOD) predictions.

Traditionally, designing a high-fidelity simulator from field data has been a craft exercise: slow and costly, requiring domain experts to translate a task brief and historical observations into a structural blueprint and to hand-tune parameters using priors and experience. This artisanal process is hard to validate and difficult to reproduce at scale. Standardization protocols for agent-based models, such as ODD (Overview, Design concepts, and Details) (Grimm et al., 2010; Müller et al., 2013; Laatabi et al., 2018; Grimm et al., 2020), improved transparency and reproducibility by introducing a common documentation framework. However, they do not reduce the dependence on expert-driven craft, nor do they resolve deeper challenges of *calibration* (systematically fitting simulators to data) and *generalization* (ensuring credible behavior under intervention).

Recent work has begun to alleviate parts of this bottleneck. Large language models (LLMs) are being used in social computing and multi-agent research; they propose and even code executable simulators that reproduce and predict complex multi-agent phenomena (Yang et al., 2024; Gao et al., 2024; Yamada et al., 2025; Tang et al., 2025; Piao et al., 2025; Zhang et al., 2025; Wang et al., 2025; Zhou et al., 2025; Li et al., 2025). In parallel, a separate line of research tackles the calibration problem directly. Because many simulators are non-differentiable—effectively black boxes with no tractable likelihoods or gradients—parameter fitting has turned to *Bayesian Optimization (BO)* or *Simulation-Based Inference (SBI)* as sample-efficient ways to search for high-fidelity parameterizations or full

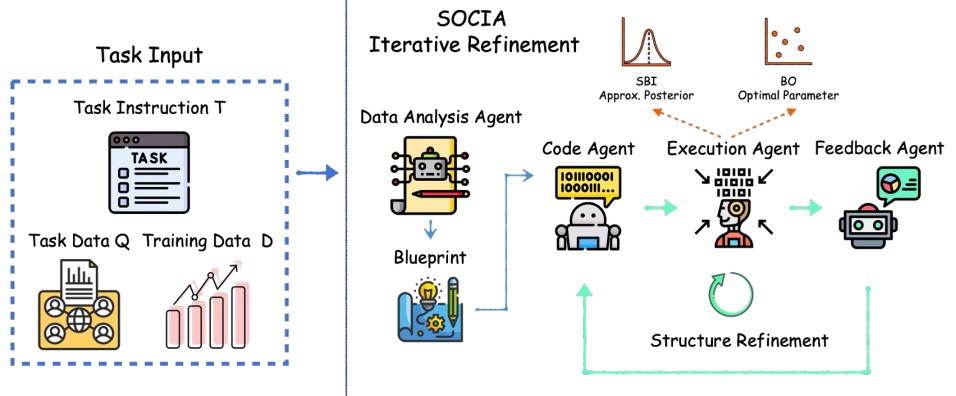

Figure 1: **SOCIA overall framework.** *Left—Task intake:* A task brief and multi-source data are analyzed by the *Data Analysis Agent* (DAA). *Center—Structure synthesis:* Guided by Chain-of-Structure (CoS), the DAA distills an executable blueprint—agents, topology, policy knobs, state/-transition rules, observables, and a calibration schema—which the *Code Generation Agent* (CGA) compiles into runnable code. *Right—Calibration & evaluation:* The *Simulation Execution Agent* (SEA) runs the simulator and calibrates via two pipelines—Bayesian Optimization and Simulation-Based Inference. Validation metrics drive the *Feedback Generation Agent* (FGA) to propose edits, forming a CGA→SEA→FGA loop until convergence.

posteriors (Gutmann & Corander, 2016; Eriksson et al., 2019; Reiker et al., 2021; McCulloch et al., 2022; Dyer et al., 2024; Holt et al., 2025).

These advances address important subproblems but remain siloed: LLM-based simulators automate structure yet typically lack principled calibration or uncertainty handling; BO/SBI methods provide powerful calibration tools for black-box models, but they operate under the assumption of a fixed structure and therefore cannot generate or revise the mechanisms themselves.

As a result, building simulators from data remains challenging along three tightly coupled axes: **(1) Structure design.** When only numeric datasets and a short task brief are available, **structure is the first-order decision**: which agent archetypes and attributes to include; who interacts with whom and how; whether interactions form explicit topologies; what exogenous/intervention signals reach agents; how agents perceive/act; and which decision policies govern behavior. These choices fix the **topology of modules and functions** and ultimately determine whether emergent dynamics match observed data and task semantics (Grimm et al., 2020). **(2) Parameter calibration.** Once a structural blueprint is set, **edge weights, rates, and thresholds** control information flow across modules. Selecting values that achieve **high-fidelity** alignment with data is non-trivial for non-differentiable, black-box simulators that lack analytic likelihoods—ruling out standard gradient/MLE procedures and motivating black-box calibration (Gutmann & Corander, 2016). **(3) Beyond in-distribution: OOD extrapolation.** Simulators are expected not only to fit historical data but also to **generalize under interventions**—the *what-if* regimes where exogenous signals or policies shift. Credible extrapolation requires mechanism-faithful structures that remain valid off the training manifold, paired with principled uncertainty handling to avoid brittle forecasts (Griesemer et al., 2024).

As shown in Figure 1, this paper addresses these three challenges with **SOCIA** (**S**imulation **O**rchestration for **C**omputational **I**ntelligence with **A**gents), a system that couples **LLM-guided structural design** with **data-driven calibration** and **extrapolation-oriented uncertainty handling**. Concretely, given a natural-language modeling brief and observational data, SOCIA automatically constructs an executable, calibrated multi-agent simulator: it (i) generates and iteratively refines simulator code, (ii) calibrates the simulator's parameters against data using black-box optimizers (BO or SBI), and (iii) returns the resulting simulator together with diagnostics and forecast trajectories. Throughout, we use *simulation agents* to refer to entities inside the modeled world (e.g., users, community residents), and *LLM orchestration agents* (DAA, CGA, SEA, FGA) to refer to the LLM-based components that construct and calibrate simulators.

- **Chain-of-Structure (CoS) specification for structure.** From the task brief and training data, we construct a data schema and pose a **stepwise, task-agnostic reasoning chain** to an LLM-based agent (the Data Analysis Agent, DAA). CoS elicits blueprint decisions—*simulation agents and their archetypes*, interaction topology, information propagation, exogenous/intervention channels, decision policies, evaluation metrics, and parameters to be calibrated—yielding a mechanism-rich structure $\lambda$ anchored in observed data rather than free-form prose.

- **Explicit parameterization and calibration loop.** We make the simulator's parameters explicit as a vector $\omega$ and recast tuning as **parameter optimization under fixed $\lambda$**. SOCIA integrates **Bayesian Optimization with Gaussian Processes** (for sample-efficient point calibration on noisy, non-differentiable objectives) and **Simulation-Based Inference (SBI)** (to learn a posterior distribution over parameters by training a conditional density estimator). An **iterative refinement outer loop** edits the structure when calibration diagnostics indicate structural inadequacy, then re-calibrates, progressively co-optimizing $\lambda$ and $\omega$.

- **Extrapolation by design plus posterior uncertainty.** The CoS specification explicitly encodes **exogenous and intervention signals** and policy-dependent agent updates, enabling controlled *what-if* manipulation at the structural level. **SBI posteriors** provide uncertainty-aware forecasts (posterior-predictive trajectories) that propagate parameter uncertainty into future scenarios, improving robustness when conditions shift. The combination—mechanism-aware structure and distributional calibration—yields simulators that are accurate where data exist and **credible under OOD interventions**.

In this work, we instantiate SOCIA primarily for agent-based and micro-simulation settings in social domains, and all notation and experiments are framed in terms of agent-based models. The core orchestration pattern is, in principle, agnostic to model class and could wrap any simulator that (i) exposes a parameterized structure representation and (ii) can be evaluated via diagnostics.

In summary, this paper makes **three contributions**: (1) it casts simulator construction as joint structure–parameter co-optimization, making explicit three deliverables—mechanism-rich structural blueprints, a tunable parameter set, and a calibration schema—thus defining an operational modeling recipe; (2) it introduces an end-to-end co-optimization pipeline that alternates targeted structural edits with parameter calibration, using pluggable black-box calibrators to efficiently fit simulators under tight budgets; and (3) it demonstrates consistent gains across tasks, with ablations establishing the necessity of structure–parameter co-optimization for high-fidelity simulator construction.

## 2 RELATED WORK

**LLM-driven automatic simulator code construction.** LLM-driven agent-based simulation has opened new opportunities, yet existing approaches face limitations in **scalability**, **accessibility**, **automation**, **validation**, and **generalizability**. For **scalability**, AgentSociety (Piao et al., 2025), OASIS (Yang et al., 2024), and SocioVerse (Zhang et al., 2025) scale to millions of agents but mainly provide environments/engines rather than simulator code, lacking parameter calibration and systematic validation. For **validation**, YuLan-OneSim (Wang et al., 2025) extends the ODD protocol (Grimm et al., 2010) for bottom-up checks, while AI Scientist-v2 (Yamada et al., 2025), GenSim (Tang et al., 2025), and AgentScope (Gao et al., 2024) focus on idea drafting, module assembly, or fault tolerance (retries, rule-based repair, semantic critiques), all without statistical calibration or cross-domain generality. For **automation/accessibility**, Sotopia-S4 (Zhou et al., 2025) enables natural-language prototyping and AgentSwift (Li et al., 2025) explores hierarchical search for LLM-agent design, but neither produces executable code or robust validation pipelines.

**Simulator Calibration with Bayesian Methods.** Work on Bayesian Optimization and Simulation-Based Inference has advanced parameter tuning but key gaps remain. Likelihood-free schemes such as Approximate Bayesian Computation are simulation-hungry and assumption-heavy (Dyer et al., 2024; Jennings & Madigan, 2017), and—while powerful—can be computationally intensive and sensitive to misspecification (Wilkinson, 2013); Bayesian conditional density estimation improves this but still struggles to scale (Papamakarios & Murray, 2016). Calibration frequently suffers from structure–model discrepancy because simulator logic is fixed (McCulloch et al., 2022). GP surrogates in BO increase efficiency and support multi-objective search (Reiker

et al., 2021), and emulator networks likewise cut likelihood-free simulation costs (Lueckmann et al., 2019), yet these typically retune numeric parameters to past data and require re-calibration in new regimes. SOCIA goes further by *co-optimizing* parameters and structure via a principled Chain-of-Structure that elevates exogenous/policy signals to first-class inputs and by exposing SBI posterior predictives for scenario extrapolation through modular interfaces. The most related system, G-SIM (Holt et al., 2025), also seeks joint optimization but relies on free-form LLM textual proposals and lacks a comparable structured protocol.

## 3 SOCIA: HYBRID SIMULATOR CONSTRUCTION AND CALIBRATION

### 3.1 PROBLEM DEFINITION

We define a *simulator* as a calibrated, executable, discrete-time dynamical program $C(\lambda, \omega, \phi)$. The triple $(\lambda, \omega, \phi)$ is the deliverable to the user:

- **structure** $\lambda$: the mechanistic blueprint of the world—*simulation-agent*/entity definitions;behavior and (if applicable) communication modalities; interaction/topology (graphs, layers); observable inputs available to agents; exogenous signals or interventions from the environment; and decision policies that map observations to actions (Holt et al., 2025).
- **parameters** $\omega$: tunable numerical parameters that govern quantitative relationships between components (weights, probabilities, etc.) (Avegliano & Sichman, 2023).
- a **calibration schema** $\phi$: the algorithmic recipe that trains and validates $(\lambda, \omega)$, i.e., data splits, objective(s), constraints/priors, optimizer or inference procedure, stopping criteria, and any iteration strategy (Werker & Brenner, 2004).

Let $x_t \in \mathcal{X}$ be the system state at time $t$, $u_t \in \mathcal{U}$ the exogenous inputs (signals or policy levers), and $\hat{y}_t \in \mathcal{Y}$ the simulator's observables. Critically, calibration relies solely on the observables $y$, as the internal state $x$ is typically inaccessible. The executable component of $C$ implements the transition:

$$x_{t+1} = F(x_t, u_t; \lambda, \omega), \qquad \hat{y}_t = H(x_t; \lambda) \tag{1}$$

Execution can be explicit (roll out $F$ step-by-step to generate trajectories) or implicit (use $F$ as a one-step predictor). After execution, $H$ will update the simulator's observables.

Beyond the temporal dimension, the simulator $C$ must also accommodate different task regimes: (a) **Descriptive/Predictive** tasks, which rely on in-distribution (ID) fitting to historical data; and (b) **What-if** intervention analysis, which requires *out-of-distribution (OOD) extrapolation* by altering the exogenous inputs $u_t$ or switching policies within $\lambda$. In other words, $C(\lambda, \omega, \phi)$ is capable of generating trajectories in the real system's state space and producing accurate state transitions even under novel OOD scenarios.

Given a training dataset $D = \{(y_t, u_t)\}_{t=1}^{T_{\text{steps}}}$, natural-language task brief $T$, and auxiliary task data $Q$ (simulation-agent attributes, behavioral logs, environmental context, etc), calibration seeks $(\lambda^*, \omega^*)$ that (i) aligns with factual observations and (ii) maintains stability and plausibility under interventions.

In this work, we use $\mathcal{A}(\cdot)$ to denote a *statistical aggregation operator* that maps micro-level individual simulation outputs (e.g., each agent's trajectories, ratings, or binary mask-wearing decisions) into macro-level indicators (e.g., daily adoption rates, aggregate star-rating distributions, trajectory summaries). These aggregates are then compared to their ground-truth counterparts for calibration and evaluation. We denote by $y_{\text{obs}}$ the observed aggregates (or summary statistics) used as conditioning variables for SBI in Section 3.3. With a dataset $D$,

$$J(\lambda, \omega; D) = \sum_{(y_t, u_t) \in D} \mathcal{L}\big(\mathcal{A}(y_t), \mathcal{A}(\hat{y}_t(\lambda, \omega))\big) + \Psi(\lambda, \omega), \tag{2}$$

where $\mathcal{L}$ is instantiated using the same task-specific discrepancy metrics reported in our experiments (e.g., aggregate RMSE/MAE, Brier score, trajectory-discrepancy measures such as TransitionFit on aggregated trajectories), and $\Psi$ encodes hard/soft domain constraints (feasibility, conservation, monotonicity, coverage). The calibration schema $\phi$ specifies how $\omega$ is optimized for a fixed $\lambda$ (e.g., Bayesian Optimization (BO) or simulation-based inference (SBI)) and how adequacy is diagnosed via validation metrics.

## 3.2 Constructing $\lambda$ and $\omega$

**From heterogeneous inputs to a unified schema $D_s$.** The simulator is *data-driven*. Beyond the training observations $D$ and the task description $T$, we incorporate task data $Q$ that (i) defines *simulation-agent* properties, (ii) characterizes the spatiotemporal environment, and/or (iii) records past simulation-agent behaviors. These heterogeneous sources are jointly analyzed into a *data schema $D_s$* enumerating: entity/agent sets and populations; observable and latent state variables (types/units/ranges); interaction topologies (graphs/layers for behavior or communication); exogenous drivers $u_t$ and intervention signals; macro-level aggregates $\mathcal{A}$ and evaluation windows.

**LLM-guided Chain-of-Structure (CoS) prompting to elicit a task blueprint $B$.** To transform $D_s$ and $T$ into an executable design, we introduce *Chain-of-Structure (CoS) specification*. Inspired by chain-of-thought prompting techniques (Wei et al., 2022; Wang et al., 2024c), CoS spec breaks down the complex design task into a sequence of focused reasoning steps, each eliciting a specific aspect of the simulator's structure or parameters. This stepwise decomposition encourages the LLM to incorporate domain context and data-derived insights systematically, resulting in a well-structured *task blueprint $B$ that specifies the simulator structure $\lambda$ and its tunable parameter set $\omega$*.

In particular, the CoS spec template leads the **Data Analysis Agent (DAA)** through (details in Appendix): (1) **Overall simulation design**: clarify the target phenomena/outcomes and modeling stance; (2) **Scale & granularity**: specify time step, spatial resolution, and population size; (3) **Agent archetypes**: define *simulation-agent* unit(s) and roles;; list static attributes and dynamic states of agents; (4) **Interaction topology**: describe how agents interact (graphs, layers, weights); (5) **Information propagation**: determine whether/how information diffuses; topology/mechanism; how inputs parameterize it; (6) **Exogenous signals**: identify external signals/interventions and their access paths to agents/modules; (7) **Action decision policy**: define actions and the policy mapping observations/signals to actions; (8) **Hold-out strategy**: propose training/validation splitting to avoid leakage and to enable OOD probes; (9) **Simulation evaluation**: define task-relevant metrics and macro aggregations $\mathcal{A}$); (10) **Calibratable parameters**: enumerate tunable parameters with bounds/priors and coupling constraints.

Through CoS, the DAA effectively generates a preliminary simulator design that is structurally grounded in domain knowledge and informed by data. Notably, this multi-step reasoning mitigates omissions and implausibilities by forcing the model to explicitly account for all elements (agents, variables, interactions, etc.) in the schema before finalizing $\lambda$. The resulting task blueprint $B$ provides an initial specification of the simulator's structure $\lambda$ and parameterization $\omega$.

## 3.3 From blueprint to $\phi$ and executable code $C$

The task blueprint $B$ serves as the primary guideline (Grimm et al., 2010; Müller et al., 2013; Laatabi et al., 2018; Grimm et al., 2020; Wang et al., 2025) for the **Code Generation Agent (CGA)** when generating executable code. Beyond this structural reference, we must also determine the calibration schema $\phi$. To this end, we inject both the blueprint $B$ and the task-specific data schema $D_s$ into the CGA — an LLM-driven *orchestration agent*. This not only directs the CGA to compile the simulator structure $\lambda$ and expose its parameter vector $\omega$, but also guides it to construct the calibration schema by: (i) designing validation and training splits; (ii) specifying the execution mode (step-by-step rollouts vs. implicit one-step predictors, reflecting the known trade-offs between single-step recursion and direct multi-step prediction in sequential models (Somalwar et al., 2025)); and most critically, (iii) embedding two complementary calibrators within the generated code—**Bayesian Optimization with Gaussian Processes (BO)** and **Simulation-Based Inference (SBI)**—to tune the parameter set $\omega$.

**Calibrators.** With structure $\lambda$ fixed, `SOCIA` frames simulator tuning as searching for parameter configurations $\omega$ that minimize a task-specific objective $\mathcal{J}(\omega)$, defined via **CoS Point 9 (Simulation Evaluation)**. We employ two complementary calibrators:

- **Gaussian-process Bayesian Optimization (GP-BO, TuRBO).** BO treats $\mathcal{J}(\omega)$ as a black-box, fits a GP surrogate, and selects the next candidate $\omega$ via an acquisition rule (e.g., EI/UCB) (Jones et al., 1998). We adopt **TuRBO (trust-region BO)** (Eriksson et al., 2019) for scalability and to capture local curvature. TuRBO+GP yields (i) high sample efficiency for expensive, non-differentiable simulators; (ii) principled treatment of noisy objectives

through Monte Carlo averaging; and (iii) adaptive trust regions that prevent overconfident global surrogates in high-dimensional parameter spaces.

- **Simulation-Based Inference (SBI: MAF + NPE).** SBI learns the *posterior* distribution over parameters, rather than a single point estimate, enabling uncertainty-aware calibration and diagnostics (Cranmer et al., 2020; Hull et al., 2024; Deistler et al., 2025). We use Masked Autoregressive Flows (MAF) (Papamakarios et al., 2017) with Neural Posterior Estimation (NPE) (Papamakarios & Murray, 2016) to approximate the posterior $p(\omega|y_{obs})$. Posterior inference provides: (i) identifiability checks (sharp vs. diffuse posteriors) that signal structural adequacy under regime shifts (Amaranath et al., 2025); (ii) posterior predictive bands that propagate parameter uncertainty into scenario forecasts, improving coverage under interventions (Sinharay & Stern, 2003; Gu et al., 2024); and (iii) robust decision support by marginalizing over plausible $\omega$ rather than committing to brittle point estimates (Falkiewicz et al., 2023).

In summary, GP-BO (TuRBO) acts as a fast, sample-efficient *point calibrator*, while SBI (MAF+NPE) functions as a *distributional calibrator* that delivers $p(\omega \mid x)$ and posterior-predictive rollouts, providing robustness for what-if and OOD extrapolation tasks. In practice, BO and SBI are instantiated as *alternative* calibration schemas rather than sequential stages: when only a point estimate is required, SOCIA runs BO and takes the BO optimum as $\omega^*$; when posterior uncertainty is needed, SOCIA instead runs SBI to approximate $p(\omega \mid D, \lambda)$ and instantiates the simulator with a draw from this posterior as $\omega^*$, while retaining the posterior for uncertainty quantification. We do not combine BO and SBI outputs within a single calibration run.

Given $B(\lambda, \omega)$, the **CGA** compiles it into an executable program $C(\lambda, \omega, \phi)$ with explicit parameters and calibration hooks (data loaders, reproducible splits, objectives/diagnostics, constraints).

## 3.4 ID FITTING VS. OOD EXTRAPOLATION

**Task regimes.** The simulation tasks considered in this paper can be broadly divided into **Descriptive/Predictive** tasks based on *ID fitting* (Holt et al., 2025) and **What-if** intervention tasks involving *OOD extrapolation* (Zhang et al., 2024), which differ substantially in nature. ID fitting asks the simulator to reproduce behavior under conditions similar to those seen in training—i.e., interpolate within the empirical regime. OOD extrapolation instead requires credible behavior under unseen conditions or interventions, often with shifted exogenous signals or altered mechanisms. The latter is not simply "harder prediction": it demands that the simulator's logic embody mechanism-level regularities that remain valid when inputs or constraints change (Silva et al., 2023).

**How CoS helps structural extrapolation.** Our CoS prompt turns simulator design into an explicit reasoning chain that equips the structure $\lambda$ with extrapolation affordances rather than mere curve-fit artifacts. Concretely: (i) **Exogenous signals (Point 6)** make "what-if" conditions first-class inputs—policy toggles and shock processes that drive agent updates (Nitzsche & Simm, 2024). (ii) **Interaction topology (Point 4)** encodes who influences whom, so novel shocks propagate plausibly across networks rather than via ad-hoc correlations. (iii) **Information propagation (Point 5)** specifies diffusion mechanisms (e.g., social influence) that generate emergent adoption even when training data lack that behavior. (iv) **Action decision policy (Point 7)** ties agent actions to risk, cues, and mandates, giving policy-sensitive rules that respond sensibly under new regimes (Avegliano & Sichman, 2023). Together, these CoS elements instantiate causal pathways and guardrails that are portable beyond the training distribution.

**Solving both with structure + calibration.** SOCIA combines mechanism-rich $\lambda$ with parameter calibration $\omega \leftarrow \varphi$ to achieve fidelity ID and credibility OOD. We first fit $\omega$ on factual data and establish empirical alignment. We then exercise the same calibrated simulator under altered $u_t$ or policy switches in $\lambda$ to test extrapolation. When validation under shifted regimes reveals bias or constraint violations, SOCIA's outer loop edits $\lambda$ (via feedback) and re-calibrates $\omega$, selecting structures that both explain the observed regime and generalize under interventions. The result is a simulator that is accurate where data exist and principled where they do not. Moreover, the posterior distributions from SBI, when coupled with structured design, provide uncertainty-aware forecasts that enhance robustness under extrapolation (Masserano et al., 2022; Saoulis et al., 2025; Alvey et al., 2025), as they propagate parameter uncertainty into future scenarios rather than relying on brittle point estimates.

## 3.5 Iterative Refinement

Once the **CGA** compiles an executable program $C(\lambda, \omega, \phi)$, the **Simulation Execution Agent (SEA)** and the **Feedback Generation Agent (FGA)** jointly implement a nested structure–parameter refinement loop.

**Inner parameter-calibration loop.** For any fixed blueprint $\lambda$ and schema $\phi$, and dataset $D$, we write the validation loss as $J(\lambda, \omega; D)$ under parameters $\omega$. Starting from $\omega_0$, the SEA runs a parameter-calibration loop under fixed $\lambda$, optimizing $\omega$ according to $\phi$ (e.g., Bayesian optimization or simulation-based inference) until convergence and returning an optimized pair $(\lambda, \omega)$ with loss $J(\lambda, \omega; D)$. Alongside this scalar loss, the SEA computes a *diagnostic vector* $\delta(\lambda, \omega)$ that bundles: (i) the overall validation loss $J(\lambda, \omega; D)$; (ii) metric-wise residuals (e.g., errors on SD/SI, DARD/STVD, sentiment/adoption rates); and (iii) time- and subgroup-resolved residuals (e.g., pre-/post-intervention phases or user cohorts). This diagnostic signal is passed to the outer loop: $J$ determines acceptance, while $\delta$ indicates where structural edits should be targeted.

**Outer structure–parameter refinement loop.** The outer loop performs greedy structure–parameter co-optimization driven by these diagnostics. We maintain a *historic log* $\mathcal{H}$ of outer iterations; each entry records the structure $\lambda$, calibrated parameters $\omega$, diagnostics $\delta(\lambda, \omega)$, scalar loss $J(\lambda, \omega; D)$, and the structural feedback proposed in that iteration. Let $(\lambda^*, \omega^*)$ denote the current best configuration with best-so-far loss $J^* = J(\lambda^*, \omega^*; D)$. At each iteration, the FGA receives as context the current $\lambda$, the corresponding simulator code, the optimized parameters $\omega$, the diagnostics $\delta(\lambda, \omega)$, and a short history $\mathcal{H}_t \subset \mathcal{H}$ of previous edits and their losses. This in-context history guides the FGA to propose a small, interpretable set of *code-level structural edits* to $\lambda$ (e.g., adding/removing mechanisms, adjusting interaction topology, re-parameterizing rates as state-dependent). The CGA applies these edits to synthesize a new simulator with structure $\lambda_{\text{new}}$. The SEA then re-runs the inner loop for $\lambda_{\text{new}}$, producing $\omega_{\text{new}}$, diagnostics $\delta(\lambda_{\text{new}}, \omega_{\text{new}})$, and loss $J_{\text{new}} = J(\lambda_{\text{new}}, \omega_{\text{new}}; D)$.

We adopt a *greedy update rule*: the new configuration $(\lambda_{\text{new}}, \omega_{\text{new}})$ is accepted only if it strictly improves the best-so-far loss, $J_{\text{new}} < J^*$. All simulations use fixed random seeds and averaged rollouts, so this zero-tolerance rule is numerically stable. Whenever an edit is accepted, we set $(\lambda^*, \omega^*) \leftarrow (\lambda_{\text{new}}, \omega_{\text{new}})$ and $J^* \leftarrow J_{\text{new}}$, making the sequence of best-so-far losses monotonically non-increasing. To avoid a global threshold $\varepsilon$, we instead use patience and a fixed budget on outer iterations: we maintain a counter of consecutive iterations with no accepted edit, and terminate once no accepted edit has improved $J^*$ for $K = 2$ consecutive iterations or a budget of 5 outer iterations is exhausted. After each iteration, the current configuration, diagnostics, loss, and proposed feedback are appended to $\mathcal{H}$ for use as in-context history in subsequent iterations.

The final output of SOCIA is the calibrated pair $(\lambda^*, \omega^*)$ and its schema $\phi$, ready for both *descriptive/predictive* use and *what-if* analyses under altered simulation conditions. Because Chain-of-Structure prompting already constrains the initial blueprint $\lambda$ to a plausible region, only a small number of diagnostic-driven, FGA-guided refinements are typically required, in contrast to the broad evolutionary search used in G-SIM (Holt et al., 2025). The overall SOCIA's framework is shown in Figure 1, and the algorithmic workflow is summarized in Algorithm 1 (Appendix A).

## 4 SOCIA: Empirical Study

### 4.1 Experiment settings

**Benchmarks.** We evaluate SOCIA on three real-world–inspired simulation tasks: (1) **User Modeling**, comes from the AgentSociety Challenge (Yan et al., 2025) and requires using LLM as a tool to predict ratings for new products based on users' historical star ratings on an e-commerce platform; (2) **Mask Adoption Behavior Simulation**, captures the temporal dynamics of mask-wearing decisions in a socially embedded population during a pandemic. Decisions are shaped by heterogeneous social ties, risk perception, and governmental interventions, with a public health campaign on Day 10 triggering diffusion. Inspired by data paradigms from BESSIE (Mortveit et al., 2022) and pandemic-era mask decision simulators (Mitsopoulos et al., 2023), we generate manually perturbed, LLM-based synthetic data for 1,000 agents embedded in a multi-relational graph; (3) **Personal Mobility Generation**, adopts the LLMob dataset (Wang et al., 2024a), which contains real-world

spatiotemporal trajectories from residents in Japan. The objective is to predict each individual's next-day mobility trajectory under three settings: (a) normal-to-normal prediction, (b) abnormal-to-abnormal (pandemic) prediction, and (c) abnormal prediction using only normal-period history.

**Task regimes.** The **ID/OOD** distinction maps to tasks as follows: **User Modeling** is predominantly *ID fitting*; **Mask Adoption** is a genuinely *OOD/intervention* problem; and **Personal Mobility** includes both regimes (normal→normal and pandemic→pandemic as *ID*; normal→pandemic as *OOD*). From an execution-mode perspective, **User Modeling** uses *implicit one-step predictors*, whereas **Personal Mobility** and **Mask Adoption** use *step-by-step rollouts* (multi-step forecasting).

**Evaluation metrics.** We report **means with 95% CIs**. Stochastic LLM- and simulator-based runs are repeated with **five seeds** (Colas et al., 2018), while **SBI inference** draws 2,000 posterior samples to propagate parameter uncertainty (Falkiewicz et al., 2023). Task-specific metrics follow **CoS**-structured prompts (Section 3.2): **MAE** for **User Modeling** (Wang et al., 2024b); **RMSE** for **Mask Adoption**, emphasizing large deviations; and **DARD/STVD** for **Personal Mobility Generation**, measuring activity–time and spatio-temporal footprint alignment. All are **error distances**, where lower ↓ indicates better ↑ performance.

**Baseline Methods.** We compare `SOCIA` against a range of baseline approaches. The first is **AI Scientist-v2** (Yamada et al., 2025), an automated pipeline that ingests experimental data, constructs and executes models or experiments, and autonomously analyzes and reports results (note that due to the complexity of simulator construction, AI Scientist-v2 still requires minor manual code adjustments to complete this task). We include two basic parameter calibrators. **Random Search** (Bergstra & Bengio, 2012) samples parameter vectors from the same prior ranges used by SOCIA, evaluates the simulator under $J_{val}$ for each sample, and reports the best-performing setting found. **LR** (Li & Lederer, 2019) is a simple surrogate-based scheme: it fits a lightweight logistic-regression model on previously evaluated parameter–loss pairs and uses this surrogate to propose the next candidates in parameter space, always tuning the same underlying simulator structure. The G-SIM family (Holt et al., 2025) serves as another key baseline: it employs LLMs to generate simulator code and integrates gradient-free calibration methods, including evolutionary strategies (**G-SIM-ES**) and simulation-based inference (**G-SIM-SBI**), to obtain executable, calibrated, and uncertainty-aware simulators. Finally, we evaluate two branches of our own framework: **SOCIA-BO**, which leverages TuRBO+GP for BO, and **SOCIA-SBI**, which applies MAF with NPE to learn SBI.

**Implementation.** All `SOCIA` agents—as well as the reproduced baseline models—are built on the GPT-5 (OpenAI, 2025). We re-implemented baselines either via code reproduction or via dataset-level reproduction to ensure evaluation consistency. We leverage BoTorch (Balandat et al., 2020), sbi (the PyTorch toolkit for simulation-based inference) (Tejero-Cantero et al., 2020), and Evo-Torch (Toklu et al., 2023), along with other PyTorch-based libraries (Paszke et al., 2019), to implement the parameter optimization algorithms used across the approaches evaluated in this work.

For SBI, unless otherwise noted, all tasks draw 2,000 posterior samples per calibrated simulator to compute predictive means and confidence intervals. For the User Modeling task, a black-box LLM yields one-step ratings and review features. We cache its outputs at the (user, item, prompt) level and reuse them across SBI: each posterior draw $\omega$ merely rescales/transforms the cached scores, so drawing 2,000 posterior samples does not require 2,000 additional LLM calls.

Across tasks, `SOCIA` typically requires 3–4 outer-loop iterations to obtain stable, high-quality simulator code: in our setup, the first draft of a simulator is produced in about 17 minutes and each refinement in about 10 minutes, for a total end-to-end optimization time of roughly 40 minutes wall-clock per task. Each iteration uses a fixed small set of LLM calls (planner / critic / editor / verifier), resulting in an overall LLM budget of approximately $10^6$ tokens per task.

### 4.2 MAIN FINDINGS

**Overall: `SOCIA` dominates.** As shown in Table 1, `SOCIA`-BO is best on *User* (MAE **0.55**±0.06) and *Traj. N→N* (DARD/STVD **0.44/0.45**); `SOCIA`-SBI is best on *Mask* (RMSE **0.11**±0.07), *Traj. A→A* (**0.46/0.50**), and the hardest *Traj. N→A* OOD (**0.52/0.58**). This pattern matches `SOCIA`'s CoS design with explicit policy/topology knobs and a built-in calibration schema (BO+SBI).

**ID vs. OOD.** *User*, *Traj. N→N*, *Traj. A→A* are ID. Here, when objectives are smooth and structure aligns with data, **BO** attains the lowest ID error (*User.* and *Traj. N→N*), while **SBI** leads *Traj. A→A*

Table 1: Evaluation results (±denotes 95% CIs) on three simulation tasks, and lower values indicate better performance. The best and second-best results are highlighted in **bold** and underlined. **Traj. N→N**, **Traj. A→A**, and **Traj. N→A** denote **Personal Mobility** (training on normal period data and predicting normal period trajectory), abnormal-to-abnormal (pandemic) prediction, and abnormal prediction using only normal-period history, respectively.

| Methods ↓ | User. | Mask. | Traj. N→N (ID) | | Traj. A→A (ID) | | Traj. N→A (OOD) | |
|---|---|---|---|---|---|---|---|---|
| | MAE ↓ | RMSE ↓ | DARD ↓ | STVD ↓ | DARD ↓ | STVD ↓ | DARD ↓ | STVD ↓ |
| AI Scientist-v2 | 0.89±0.10 | 0.45±0.07 | 0.77±0.06 | 0.82±0.06 | 0.78±0.06 | 0.83±0.06 | 0.80±0.07 | 0.85±0.06 |
| Random Search | 0.86±0.10 | 0.27±0.09 | 0.70±0.05 | 0.78±0.06 | 0.74±0.06 | 0.80±0.06 | 0.77±0.07 | 0.83±0.06 |
| LR | 0.78±0.09 | 0.32±0.06 | 0.59±0.04 | 0.60±0.04 | 0.60±0.05 | 0.63±0.05 | 0.72±0.06 | 0.79±0.06 |
| G-SIM-ES | 0.58±0.07 | 0.37±0.08 | 0.45±0.02 | 0.48±0.03 | 0.52±0.04 | 0.56±0.04 | 0.58±0.05 | 0.64±0.05 |
| G-SIM-SBI | 0.69±0.08 | 0.20±0.10 | 0.54±0.03 | 0.56±0.03 | 0.47±0.03 | 0.53±0.03 | 0.60±0.05 | 0.66±0.05 |
| SOCIA-BO | **0.55±0.06** | 0.24±0.09 | **0.44±0.02** | **0.45±0.03** | 0.54±0.04 | 0.58±0.04 | 0.54±0.05 | 0.60±0.05 |
| SOCIA-SBI | 0.63±0.04 | **0.11±0.07** | 0.50±0.03 | 0.52±0.03 | **0.46±0.03** | **0.50±0.03** | **0.52±0.05** | **0.58±0.05** |

by propagating parameter uncertainty within the same regime. *Mask* and *Traj. N→A* are what-if/OOD. Because intervention/broadcast diffusion is *explicitly* encoded in structure, **SBI** learns a well-identified posterior over those levers and achieves the best OOD scores on both tasks; BO is a close second on *Traj. N→A*.

**BO vs. SBI vs. ES. SBI < BO** on some ID settings that favor efficient point search (*User.* and *Traj. N→N*), but **SBI > BO** on OOD/structured-intervention tasks (*Mask*, *Traj. N→A*) and even on ID setting *Traj. A→A*. Compared to **ES**, **BO** attains lower errors (e.g., *Traj. N→N*, *Traj. N→A*) via surrogate + trust-region efficiency under noisy, weakly identifiable parameters; **ES** is competitive (often runner-up on *User.* and *Traj. N→N*) but less sample-efficient.

*G-SIM* and *AI Scientist*: *structural gaps*. G-SIM couples LLM proposals with gradient-free calibration (ES/SBI) but relies on a rigid NumPy-style template biased toward compartmental epidemiology, lacks explicit structural prompting, and omits task-specific heads for feature-conditioned, ordinal/trajectory outputs—hence G-SIM-ES/SBI lag SOCIA across columns. AI Scientist-v2 automates generic ML pipelines rather than domain simulators; without user–item conditioning, stochastic calibration, or uncertainty-aware evaluation, it performs poorly on all tasks.

**Why LR/Random are weak?** **LR** cannot represent nonlinear, networked, intervention-coupled mechanisms; **Random Search** burns budget in high-dimensional, weakly identifiable spaces—neither yields calibrated, structure-aligned fits.

### 4.3 Insight gained from the design of the simulator structure

In this section, our goal is to provide *qualitative case studies* built from controlled modifications of our main tasks, in order to showcase specific capabilities of SOCIA's generated simulators.

**User modeling: black-box & one-step predictor.** In this task, the simulator (see Appendix E for design details) is recommended to call an LLM to produce a rating, but the LLM is a **black box**—*non-differentiable and without an analytic likelihood*—so gradient descent and MLE are inapplicable. Instead, *BO/EA* can directly search parameters against a chosen loss, and *SBI* learns a posterior over parameters under likelihood-free conditions. The task is also an **implicit one-step predictor** (no rollouts), so observations can only be constructed from user/item history at a single step. SOCIA addresses both issues by: (i) prompting the LLM with "user history + target-item history" to get a raw score, and (ii) passing it through a light, interpretable calibration head with a small parameter set (*affine correction, feature fusion weight, user/item biases, noise scale*). These parameters are then tuned via BO (point estimates) or inferred via SBI (posterior with uncertainty), yielding calibrated, uncertainty-aware predictions despite the black-box and one-step constraints.

**Mask Adoption: Exogenous signal.** In this simulator (Appendix F), a government intervention switches on at day 10 as an exogenous broadcast that raises public awareness; this signal then diffuses over a multilayer social network (family, work, community), where exposure accumulates via neighbor influence. Agents combine recent exposure with prior behavior, peer signals, risk perception, and demographics to decide whether to adopt masks. Because the intervention is encoded by a small, interpretable set of knobs (broadcast strength, layer transmission, memory/peer weights) that drive informative aggregate curves, the calibration becomes a well-posed likelihood-free problem—hence **SBI** excels by learning a coherent posterior and propagating uncertainty to predictions.

**Personal Mobility: ID & OOD.** In this task, `SOCIA` built the simulator in this way (Appendix G): trajectories arise from a two–stage choice (activity type $\rightarrow$ destination) with regime modifiers (closures/avoidance, mobility scaling, curfew, remote work) acting as explicit structural "knobs." When training and testing both sit in the *normal* regime (N$\rightarrow$N), the discrepancy objectives behind DARD-/STVD are locally smooth in the core choice/dwell parameters and only weakly constrained by regime knobs; this yields a well-behaved, low–noise landscape where surrogate-based **BO** can rapidly locate strong point estimates, hence the best ID error. By contrast, in *pandemic* regimes (A$\rightarrow$A) and especially under the *normal$\rightarrow$pandemic* switch (N$\rightarrow$A), the effective surface becomes rugged due to discrete availability changes, hard constraints (curfews/closures), and strong couplings between activity and destination; performance now depends on correctly *identifying* regime multipliers and propagating their uncertainty into trajectory histograms. Here **SBI** prevails: it learns a posterior over the structured knobs using trajectory summaries, averages posterior-predictive trajectories rather than committing to a single point, and leverages weakly informative priors plus policy-switched replays to remain stable under shift.

### 4.4 ABLATION STUDY

Starting from the `SOCIA`-BO backbone, we ablate five design choices: `SOCIA`-no-CoS (remove CoS specification; use a minimal task-analysis prompt), `SOCIA`-fix-$\lambda$ (disable iterative structural optimization), `SOCIA`-fix-$\phi$ (replace CoS-derived calibration design with ICL prompting), `SOCIA`-no-calib (skip parameter tuning; let the LLM suggest edits after building a simulator with random parameters), and `SOCIA`-ES (swap BO with an evolutionary calibrator). As shown in Table 2, these ablations yield the following observations.

Table 2: On the **Mask Adoption** task, we performed an ablation study with `SOCIA`-BO as the baseline. Larger positive $\Delta$RMSE indicates greater degradation.

| Models | $\Delta$**RMSE**$\downarrow$ |
|---|---|
| `SOCIA`-BO | - |
| `SOCIA`-no-CoS | $+0.17 \pm 0.06$ |
| `SOCIA`-fix-$\lambda$ | $+0.04 \pm 0.02$ |
| `SOCIA`-fix-$\phi$ | $+0.12 \pm 0.04$ |
| `SOCIA`-no-calib | $+0.20 \pm 0.07$ |
| `SOCIA`-ES | $+0.09 \pm 0.03$ |

**Ablation findings.** Using `SOCIA`-BO as the backbone, we see near-monotone degradation as we weaken structure and calibration. First, comparing **SOCIA-no-CoS** and **SOCIA-fix-$\lambda$** isolates the effect of CoS. Removing CoS in **SOCIA-no-CoS** ($+0.17\pm0.06$) forces the outer loop to search over almost unstructured simulator designs; within a hard budget of 5 outer iterations it often fails to find a good blueprint, leading to a large performance drop. In contrast, **SOCIA-fix-$\lambda$** ($+0.04\pm0.02$) keeps the CoS-guided blueprint but disables structural refinement, yielding only a small degradation: CoS is thus not an oracle but a strong, domain-informed prior that places the search in a favorable region of structure space. The largest loss appears in **SOCIA-no-calib** ($+0.20\pm0.07$), where skipping calibration leaves even good structures systematically mis-scaled, showing that structure alone is insufficient. Fixing the calibration schema in **SOCIA-fix-$\phi$** ($+0.12\pm0.04$) further shows that replacing CoS-informed calibration with generic ICL heuristics weakens inductive bias and harms identifiability, even under similar high-level structures. Finally, swapping BO for an evolutionary optimizer in **SOCIA-ES** ($+0.09\pm0.03$) confirms that ES is competitive but less sample-efficient under noisy, weakly identifiable parameters, underscoring the benefit of our BO-based inner loop.

## 5 CONCLUSION

This work presents `SOCIA`, an agentic framework that reconceptualizes simulator construction as a joint structure–parameter optimization problem. Guided by the Chain-of-Structure protocol, `SOCIA` systematically generates mechanism-rich blueprints, exposes explicit parameters, and designs calibration schemas with pluggable Bayesian Optimization and Simulation-Based Inference. An iterative refinement loop further integrates structural editing and uncertainty-aware evaluation to ensure both fidelity on observed data and credibility under interventions. Across diverse tasks—user modeling, mask adoption, and personal mobility—`SOCIA` consistently outperforms baselines, with both BO and SBI demonstrating strong performance in in-distribution fitting as well as under challenging distribution shifts. Ablation studies confirm that weakening structural design or calibration leads to predictable degradations, underscoring the necessity of `SOCIA`'s integrated approach. Together, these results establish `SOCIA` as a generalizable, reproducible, and robust paradigm for automated simulation construction.

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

**Appendix: Table of Contents**

# A    DETAILED WORKFLOW OF THE SOCIA SYSTEM

---

**Algorithm 1** SOCIA Pipeline with Structural–Parameter Calibration

---

1: **Inputs:** Full dataset $D$, task description $T$, auxiliary task data $Q$, Chain-of-Structure hints CoS
2: **Outputs:** best.$\lambda$, best.$\omega$

3: **Initialization**
4: $D_s \leftarrow$ BUILDDATASCHEMA$(D, T, Q)$
5: $B \leftarrow$ DAA$(D_s, T, \text{CoS})$
6: best $\leftarrow \{J_{\text{val}} := +\infty, \lambda := \text{None}, \omega := \text{None}, \phi := \text{None}\}$
7: no_improve $\leftarrow 0$
8: feedback $:=$ None
9: $\mathcal{H} \leftarrow \emptyset$                                             ▷ *historic log of outer iterations*
10: $t \leftarrow 0$

11: **Structural–Parameter Outer Loop**
12: **while** $t <$ max_outer_iters **do**
13:     $t \leftarrow t + 1$
14:     $C(\lambda, \omega, \phi) \leftarrow$ CGA$(B, D_s, \text{feedback})$ ▷ *CGA = Code Generation Agent; generates C from blueprint B, data schema $D_s$, and previous feedback (if any)*
15:     $(\hat{\omega}, J_{\text{val}}, \delta, \text{adequate}) \leftarrow$ SEA$(C, D)$ ▷ *SEA = Simulation Execution Agent; calibrates $\omega$ and computes diagnostic vector $\delta$*
16:     accepted $\leftarrow$ false
17:     **if** $J_{\text{val}} <$ best.$J_{\text{val}}$ **then**
18:         best $\leftarrow \{J_{\text{val}}, \lambda, \hat{\omega}, \phi\}$
19:         no_improve $\leftarrow 0$
20:         accepted $\leftarrow$ true
21:     **else**
22:         no_improve $\leftarrow$ no_improve $+ 1$
23:     feedback $\leftarrow$ FGA$(J_{\text{val}}, \lambda, \hat{\omega}, \phi, \delta, \mathcal{H})$ ▷ *FGA = Feedback Generation Agent; uses diagnostics and a short history from $\mathcal{H}$*
24:     $\mathcal{H} \leftarrow \mathcal{H} \cup \{(\lambda, \hat{\omega}, \delta, J_{\text{val}}, \text{feedback}, \text{accepted})\}$     ▷ *append current iteration to historic log*
25:     **if** (adequate $==$ true) or (no_improve $\geq 2$) **then**
26:         **return** (best.$\lambda$, best.$\omega$)
27: **return** (best.$\lambda$, best.$\omega$)

28: **Auxiliary Routines (signatures)**
29: BUILDDATASCHEMA$(D, T, Q) \rightarrow D_s$: analyzes the data schema $D_s$ based on the task description $T$, task-specific data, and the training dataset $Q$;
30: DAA$(D_s, T, \text{CoS}) \rightarrow B$: given the data schema and the Chain-of-Structure (CoS), Data Analysis Agent (DAA) guides the LLM to construct a blueprint $B$ with mechanisms, topologies, and outputs;
31: CGA$(B, D_s, \text{feedback}) \rightarrow C(\lambda, \omega, \phi)$: Code Generation Agent (CGA) generates an executable simulator $C$ according to the blueprint $B$, data schema $D_s$, and the previous iteration's feedback (if available); here, $\lambda$ denotes the simulator structure, $\omega$ the inter-structural parameters, and $\phi$ the calibration schema;
32: SEA$(C, D) \rightarrow (\hat{\omega}, J_{\text{val}}, \delta, \text{adequate})$: Simulation Execution Agent (SEA) calibrates $\omega$ until convergence, returning the calibrated parameters $\hat{\omega}$, the scalar validation loss $J_{\text{val}}$, a diagnostic vector $\delta$ (metric-wise and time-/group-resolved residuals), and an optional adequacy flag;
33: FGA$(J_{\text{val}}, \lambda, \hat{\omega}, \phi, \delta, \mathcal{H}) \rightarrow$ feedback: Feedback Generation Agent (FGA) proposes structural modification feedback based on the validation loss, structural design $\lambda$, calibrated parameters $\hat{\omega}$, calibration schema $\phi$, the diagnostic vector $\delta$, and a short in-context history extracted from the historic log $\mathcal{H}$.

---

The **SOCIA pipeline** goes beyond generating a static simulator. Instead, it constructs a complete *trainable and calibratable workflow* that can iteratively refine its own design. The core idea is to separate and coordinate three fundamental elements:

- **Structure** ($\lambda$): derived from the Chain-of-Structure (CoS) hint, specifying which mechanisms exist, how agents interact, and through which information channels signals propagate.

- **Parameters** ($\omega$): numerical quantities governing the mechanisms, such as weights, probabilities, or transition rates. These are the targets of calibration.

- **Calibration schema** ($\phi$): the training strategy that dictates how $\lambda$ and $\omega$ are combined during optimization. $\phi$ also determines dataset splits (train/validation), learning schedules, and convergence criteria.

The workflow is designed around two **nested loops**: an inner loop for parameter calibration and an outer loop for structure adaptation.

PARAMETER CALIBRATION LOOP (INNER LOOP)

Given a fixed simulator structure $\lambda$ and schema $\phi$, SOCIA runs the **Simulation Execution Agent (SEA)** to calibrate the parameters $\omega$ against the dataset $D$. The SEA executes the calibration schema $\phi$, iteratively adjusting parameters until convergence. This produces a calibrated simulator $(\lambda, \hat{\omega})$ and an associated validation loss

$$J(\lambda, \hat{\omega}; D),$$

which aggregates task-specific discrepancies (e.g., MAE/RMSE and trajectory-based statistics such as SD/SI/DARD/STVD).

In addition to the scalar loss, the SEA computes a *diagnostic vector* $\delta(\lambda, \hat{\omega})$ that exposes where and how the simulator misfits the data. Concretely, $\delta$ bundles: (i) the overall scalar loss $J(\lambda, \hat{\omega}; D)$; (ii) metric-wise residuals for each evaluation component; and (iii) time- and subgroup-resolved residuals, such as errors stratified by temporal phases (e.g., pre-/post-intervention windows) or by user cohorts (e.g., degree groups in mask adoption or mobility regimes). The SEA may also return an adequacy flag indicating whether the current simulator already meets task-specific coverage or calibration requirements. Both $J$ and $\delta$ are passed to the outer loop: $J$ is used for greedy acceptance decisions, while $\delta$ provides structured evidence that will guide targeted structural edits.

STRUCTURAL–PARAMETER OUTER LOOP

The outer loop performs diagnostic-driven structure–parameter co-optimization. SOCIA maintains a *historic log* $\mathcal{H}$ of outer iterations, where each entry records the current structure $\lambda$, calibrated parameters $\hat{\omega}$, diagnostic vector $\delta(\lambda, \hat{\omega})$, scalar loss $J(\lambda, \hat{\omega}; D)$, and the structural feedback proposed in that iteration.

Let $(\lambda^*, \omega^*)$ denote the current best configuration with best-so-far validation loss $J^* = J(\lambda^*, \omega^*; D)$. At each outer iteration, the **Feedback Generation Agent (FGA)** receives as context: the current blueprint and simulator structure $\lambda$, the calibrated parameters $\hat{\omega}$, the calibration schema $\phi$, the diagnostic vector $\delta(\lambda, \hat{\omega})$, and a short history extracted from the log $\mathcal{H}$ (previous structures, diagnostics, losses, and feedback). This history is used as in-context evidence about which kinds of edits have previously improved or failed to improve $J$, so that the FGA does not propose edits in isolation but is guided by past successes and failures.

Based on this information, the FGA proposes a small, interpretable set of *code-level structural edits* to $\lambda$—such as introducing or removing mechanisms, adjusting interaction topologies, or reparameterizing rates as state-dependent. These suggestions are passed to the **Code Generation Agent (CGA)**, which applies them to the blueprint and synthesizes a new simulator $C(\lambda', \omega', \phi)$. The pipeline then re-enters the parameter calibration loop with the modified structure $\lambda'$, and the SEA re-calibrates parameters to obtain $(\lambda', \hat{\omega}')$, diagnostics $\delta(\lambda', \hat{\omega}')$, and a new validation loss $J(\lambda', \hat{\omega}'; D)$.

We adopt a *greedy update rule*: the new configuration $(\lambda', \hat{\omega}')$ is accepted as the new best configuration only if it strictly improves the best-so-far loss,

$$J(\lambda', \hat{\omega}'; D) < J^*.$$

All simulations are run under fixed random seeds and averaged over multiple rollouts, making this zero-tolerance strict-improvement rule numerically stable. Whenever an edit is accepted, we update

$(\lambda^*, \omega^*) \leftarrow (\lambda', \hat{\omega}')$ and $J^* \leftarrow J(\lambda', \hat{\omega}'; D)$, and reset the outer-loop "no-improvement" counter. If the loss does not improve, the counter is incremented. After each outer iteration, regardless of acceptance, the current $(\lambda, \hat{\omega}, \delta, J)$ and the proposed feedback are appended to the historic log $\mathcal{H}$, so that subsequent iterations can reuse this expanded log as in-context history.

CONTROL CONDITIONS AND STOPPING CRITERIA

SOCIA maintains a record of the **historical best configuration** $(\lambda^*, \omega^*, \phi)$ with the lowest validation error $J^*$. To ensure stability and avoid unbounded search, the outer loop is controlled by two conditions:

- **Adequacy:** if the SEA flags the current simulator as adequate (e.g., prediction intervals satisfy coverage or calibration criteria), the outer loop terminates and $(\lambda^*, \omega^*)$ is returned.

- **Patience and budget:** otherwise, the loop continues until either (i) two consecutive outer iterations fail to yield any strictly better configuration (no accepted structural edit improves $J^*$), or (ii) a small fixed budget of outer iterations (5 in our experiments) is exhausted.

Under this design, the sequence of best-so-far losses $J^*$ is monotonically non-increasing by construction, and the structure–parameter refinement loop is guaranteed to terminate in a finite number of steps while adaptively stopping once further edits cease to yield meaningful diagnostic improvements.

WORKFLOW SUMMARY

The complete process is outlined in Algorithm 1. In summary:

1. **Initialization:** The **Data Analysis Agent (DAA)** analyzes the dataset $D$ and task description $T$, producing a blueprint $B$ that encodes mechanisms, topologies, and output observables, as well as a data schema $D_s$.

2. **Simulator Construction:** The **CGA** converts the blueprint $B$ (and optional feedback from prior iterations) into an executable simulator $C(\lambda, \omega, \phi)$, instantiating an initial structure $\lambda$, parameter vector $\omega$, and calibration schema $\phi$.

3. **Calibration and Diagnostics:** The **SEA** runs the calibration schema $\phi$ under fixed $\lambda$, optimizing parameters $\omega$ until convergence and reporting a scalar validation loss $J_{\mathrm{val}} = J(\lambda, \hat{\omega}; D)$ together with a diagnostic vector $\delta(\lambda, \hat{\omega})$ and an adequacy flag.

4. **Feedback and Logging:** The outer loop maintains the historical best configuration based on strict improvement in $J_{\mathrm{val}}$. The **FGA** then generates structural feedback using $(\lambda, \hat{\omega}, \phi, \delta)$ and a short in-context history extracted from the log $\mathcal{H}$, and the resulting entry $(\lambda, \hat{\omega}, \delta, J_{\mathrm{val}}, \text{feedback})$ is appended to $\mathcal{H}$.

5. **Termination:** The loop continues with updated simulators and recalibration until either adequacy is achieved or the patience/budget conditions are met (no accepted improvements for two consecutive outer iterations or exhaustion of the outer-iteration budget), at which point the best configuration $(\lambda^*, \omega^*)$ is returned.

Through this hierarchical workflow, SOCIA is capable of dynamically generating, calibrating, and validating simulators under both in-distribution and out-of-distribution (*what-if*) scenarios, while making the refinement process explicitly driven by diagnostics and constrained by principled stopping rules.

## B    DESIGN OF THE DATA ANALYSIS AGENT

In this section, we present a detailed explanation of the underlying mechanism of SOCIA's **D**ata **A**nalysis **A**gent (**DAA**). We further elaborate on how the **DAA** conducts multidimensional relational mining to uncover latent interaction patterns among entities.

1. **Key Features**

   **Input Processing Paradigm**

   - **Agent**: Takes `task_description` and optional `task_data`, building the task specification internally during processing.

   **Workflow Integration**

   - **Agent**: Combines task understanding and data analysis in a single step, returning a complete task specification with embedded analysis results.

   **Template and Prompt Structure**

   - **Agent**: Uses `templates/data_analysis_prompt.txt` with comprehensive simulation design framework.

2. **Workflow Analysis**

   **Core Process Method**

   The agent follows this workflow:

```python
def process(
    self,
    task_description: str,
    task_data: Optional[Dict[str, Any]] = None,
    mode: str = "full",
    blueprint: Optional[Any] = None
) -> Dict[str, Any]:
```

   **Task Specification Building**

   The agent first constructs a comprehensive task specification:

   *From Task Data (JSON)*

   - Extracts `task_objective` with description and simulation focus.
   - Processes `data_folder` and `data_files` specifications.
   - Incorporates `evaluation_metrics` from the task file.
   - Creates structured task specification with all necessary components.

   *From Task Description (String)*

   - Creates minimal task specification when no JSON data is provided.
   - Uses the description as the primary task objective.
   - Sets default empty values for data files and evaluation metrics.

   **Data Path Validation and File Discovery**

   - Validates data path existence (raises `FileNotFoundError` if missing).
   - Lists available files in the data directory using `_list_available_files()`.
   - Supports multiple file types: CSV, JSON, GeoJSON, pickle, and Python files.
   - Selects files for analysis based on task specification requirements.

   **Schema Inference and Semantic Analysis**

   The agent implements sophisticated schema inference:

   *CSV Schema Inference*

   - Analyzes file structure using pandas.
   - Captures data types, row/column counts, and sample data.
   - Generates comprehensive data info using `df.info()`.
   - Stores statistical summaries for numeric columns.

   *JSON Schema Inference*

- Handles both object and array structures.
- Analyzes data length and sample content.
- Captures value/item structure information.
- Provides type information for nested elements.

*Semantic Summary Generation*

- Uses LLM to generate semantic metadata for each file.
- Contextualizes file content within the simulation task.
- Stores summaries directly in schema objects.
- Provides insights on how data should inform simulation entities.

**Comprehensive Analysis Prompt**

The agent builds a sophisticated analysis prompt covering ten key areas:

(a) **Overall Simulation Design**: Objective, initialization, execution, and outputs
(b) **Scale & Granularity**: Time steps, spatial resolution, population size
(c) **Agent Archetypes**: Unit definitions, roles, attributes, and states
(d) **Interaction Topology**: Graph structures, layers, and protocols
(e) **Information Propagation**: Diffusion mechanisms and parameters
(f) **Exogenous Signals**: External interventions and effects
(g) **Action Decision Policy**: Role-specific decision functions
(h) **Holdout Plan**: Training/validation data splitting strategy
(i) **Simulation Evaluation**: Metrics and comparison methods
(j) **Calibratable Parameters**: Tunable parameters with bounds

**LLM Analysis and Response Parsing**

- Calls LLM with comprehensive analysis prompt.
- Parses JSON response into structured format.
- Transforms response to match expected data analysis result structure.
- Handles parsing errors gracefully with fallback structures.

3. **Specialized Features**

   **Blueprint Integration**

   - Designed to work with blueprint-based workflows.
   - Supports blueprint mode processing.
   - Integrates seamlessly with the code generation agent.

   **File Type Support**

   - **CSV Files**: Full statistical analysis with pandas integration.
   - **JSON Files**: Structure analysis with type inference.
   - **GeoJSON Files**: Special handling for geospatial data.
   - **Pickle Files**: Network data and preprocessed objects.
   - **Python Files**: Code analysis capabilities.

   **Semantic Metadata Generation**

   - LLM-powered semantic analysis of file content.
   - Context-aware summaries based on task description.
   - Integration guidance for simulation entities.
   - Relationship identification between data elements.

   **Comprehensive Output Structure**

   The agent returns a complete task specification with:

   - `data_analysis_result`: Structured analysis covering all ten areas
   - `file_summaries`: Formatted file information with semantic summaries
   - `schemas`: Detailed file schemas with metadata
   - Original task specification components

4. **Configuration and Usage**

   **Configuration**

```yaml
data_analysis:
  enabled: true
  prompt_template: "templates/data_analysis_prompt.txt"
  output_format: "json"
  supported_data_types: ["csv", "json", "geojson", "shapefile"]
```

   **Dependency Injection**

   - Registered in `AgentContainer` as `data_analysis_agent`.
   - Available through workflow manager agent providers.
   - Configured with output path for result persistence.

   **Integration Points**

   - Works with the task understanding agent.
   - Integrates with the code generation agent.
   - Supports blueprint-based workflows.
   - Used in test workflows for data analysis and code generation.

5. **Technical Implementation Details**

   **Error Handling**

   - Comprehensive file validation with clear error messages.
   - Graceful handling of missing files and parsing errors.
   - Fallback structures for LLM response parsing failures.
   - Detailed logging for debugging and monitoring.

   **Data Processing**

   - Efficient file scanning with type detection.
   - Memory-conscious data loading for large files.
   - Statistical analysis with pandas integration.
   - Schema inference without loading entire datasets.

   **LLM Integration**

   - Structured prompt building with template integration.
   - JSON response parsing with error recovery.
   - Semantic analysis for contextual understanding.
   - Fallback handling for malformed responses.

6. **Use Cases and Applications**

   **Blueprint-Based Simulation Development**

   - Ideal for structured simulation design workflows.
   - Supports complex multi-agent system analysis.
   - Handles calibration and validation data preparation.

   **Comprehensive Data Analysis**

   - Provides end-to-end analysis from task description to simulation design.
   - Combines data understanding with simulation parameter extraction.
   - Supports multiple file formats and data structures.

   **Iterative Development**

   - Integrates with feedback-driven workflows.
   - Maintains consistency across development iterations.
   - Supports blueprint updates and modifications.

7. **Summary**

   The **Data Analysis Agent** represents a sophisticated, blueprint-focused approach to data analysis within the SOCIA system. Its key distinguishing features include:

(a) **Integrated task specification building** that combines task understanding and data analysis

(b) **Advanced schema inference** with semantic analysis using LLM capabilities

(c) **Comprehensive simulation design framework** covering all aspects of model construction

(d) **Blueprint-based workflow integration** optimized for structured simulation development

(e) **Multi-format data support** with specialized handling for various file types

(f) **End-to-end analysis** from raw data to simulation parameter extraction

This agent serves as an alternative to the regular data analysis agent, providing enhanced capabilities for complex simulation scenarios that require comprehensive data understanding, structured analysis, and seamless integration with blueprint-based development workflows. It is particularly well-suited for scenarios where data analysis needs to be tightly coupled with simulation design and parameter calibration processes.

# C   DESIGN OF THE CODE GENERATION AGENT

In this section, we describe the **C**ode **G**eneration **A**gent (**CGA**) workflow in the `SOCIA` framework.

The **CGA** is responsible for transforming conceptual model plans into executable `Python` code for simulations. Its operation can be broken down into two major phases, i.e., *Initial Code Generation* phase and *Self-Checking and Improvement Loop* phase.

1. **Key Features**

   **Template and Prompt Structure**

   - **Agent**: Uses `templates/code_generation_prompt.txt` with a blueprint-focused approach, emphasizing production-grade code generation.

   **Input Processing**

   - **Agent**: Focuses on blueprint-based code generation with simplified input handling (commented out data source override logic).

   **LLM Interaction**

   - **Agent**: Uses `reasoning={"effort":  "high"}` for more sophisticated code generation.

2. **Workflow Analysis**

   **Core Process Method**

   The agent follows this workflow:

```python
def process(
    self,
    task_spec: Dict[str, Any],
    model_plan: Optional[Dict[str, Any]] = None,
    data_analysis: Optional[Dict[str, Any]] = None,
    feedback: Optional[Dict[str, Any]] = None,
    data_path: Optional[str] = None,
    previous_code: Optional[Dict[str, str]] = None,
    historical_fix_log: Optional[Dict[str, Any]] = None,
    mode: str = "full",
    selfloop: int = 3,
    blueprint: Optional[Any] = None
) -> Dict[str, Any]:
```

   **Prompt Building Process**

   The agent builds prompts differently based on mode:

   - **Lite Mode**: Uses minimal template with task specification, feedback, and previous code.
   - **Full Mode**: Uses comprehensive template with blueprint, model plan, data analysis, and all context.

   **Self-Checking Loop Mechanism**

   The agent implements a sophisticated self-checking system with up to 3 iterations:

   *Code Quality Check*

   - Syntax errors and runtime issues
   - Placeholder functions and missing implementations
   - Inconsistencies between model plan and implementation
   - Undefined references and poor error handling
   - Missing documentation and type annotations
   - Algorithm efficiency issues

   *Feedback Implementation Check*

   - Validates that all required fixes from previous iterations are implemented.
   - Prioritizes user feedback over system-generated feedback.
   - Checks for critical issues and code improvements.

*Historical Issues Check*

- Prevents repetition of previously fixed problems.
- Uses historical fix logs to avoid regression.
- Maintains consistency across iterations.

**Code Improvement Process**

When issues are found, the agent:

(a) Collects fixed log references from historical data.
(b) Builds improvement prompts with specific issue details.
(c) Calls LLM with high-effort reasoning for code enhancement.
(d) Validates syntax and fixes unclosed docstrings.
(e) Ensures proper entry points for execution.

3. **Specialized Features**

**Blueprint Integration**

- Designed to work with blueprint-based workflows.
- Updates blueprint with generated code metadata.
- Supports blueprint mode execution.

**Entry Point Management**

- Automatically ensures proper `main()` function structure.
- Removes traditional `if __name__ == "__main__"` guards.
- Adds direct `main()` calls for sandbox compatibility.

**Code Quality Assurance**

- Automatic detection and fixing of unclosed docstrings.
- Markdown fence removal.
- Syntax validation with auto-fix capabilities.

**Mask Adoption Patch**

- Special handling for mask-wearing simulations.
- Implements temporal holdout requirements.
- Ensures proper data splitting by unique days.

4. **Configuration and Usage**

**Configuration**

```
code_generation:
  enabled: true
  prompt_template: "templates/code_generation_prompt.txt"
  output_format: "python"
  code_style: "pep8"
```

**Dependency Injection**

The agent is registered in the `AgentContainer` as `code_generation_agent` and can be accessed through the workflow manager.

**Integration Points**

- Works with blueprint-based workflows.
- Integrates with feedback generation and iteration control.
- Supports both standalone and orchestrated execution modes.

5. **Technical Implementation Details**

**Inheritance Structure**

- Inherits from `BaseAgent` (same as regular agent).
- Shares common LLM interaction and prompt building capabilities.
- Overrides specific methods for blueprint-focused behavior.

**Error Handling**

- Comprehensive exception handling for LLM responses.
- Graceful fallback for missing templates.
- Robust JSON parsing with error recovery.

**Memory Management**

- Maintains code memory across iterations.
- Stores historical fix logs for regression prevention.
- Manages blueprint state updates.

6. **Use Cases and Applications**

   **Blueprint-Based Simulations**

   - Ideal for structured simulation development.
   - Supports complex multi-agent systems.
   - Handles calibration and validation workflows.

   **Production Code Generation**

   - Focuses on executable, production-grade code.
   - Implements comprehensive error handling.
   - Ensures deterministic behavior.

   **Iterative Improvement**

   - Supports feedback-driven development.
   - Maintains consistency across iterations.
   - Prevents regression through historical tracking.

7. **Summary**

   The **Code Generation Agent** represents a specialized, blueprint-focused approach to simulation code generation within the SOCIA system. Its key distinguishing features include:

   (a) **Blueprint-centric design** for structured simulation development.
   (b) **Enhanced self-checking mechanisms** with multi-layered validation.
   (c) **Production-grade code generation** with comprehensive error handling.
   (d) **Sophisticated feedback integration** that prevents regression.
   (e) **Specialized template system** optimized for specific simulation types.

   This agent serves as an alternative to the regular code generation agent, providing enhanced capabilities for complex simulation scenarios that require structured, iterative development with high-quality, production-ready code output.

# D  DESIGN OF THE FEEDBACK GENERATION AGENT

In this section, we describe the **F**eedback **G**eneration **A**gent (**FGA**) workflow in the `SOCIA` framework.

The **FGA** serves as the critical evaluative component in the `SOCIA` framework's iterative improvement cycle. It synthesizes information from multiple sources to provide structured, actionable feedback for improving simulation code. Here's how it works:

**1. Setup and Initialization.**   The agent begins with initialization that establishes its fundamental environment:

```python
def __init__(self, config: Dict[str, Any] = None, output_path: Optional[
    str] = None):
    # Set default config if none provided
    if config is None:
        config = {
            "prompt_template": "templates/feedback_generation_prompt.txt"
            ,
            "output_format": "json"
        }

    super().__init__(config)

    # Set output path
    self.output_path = output_path or os.getcwd()

    # Set template directory path
    self.template_dir = os.path.join(
        os.path.dirname(os.path.dirname(os.path.dirname(os.path.abspath(
            __file__)))),
        "templates"
    )
```

This setup is critical as it points to the feedback generation prompt template and establishes the directory structure for related operations.

**2. Core Feedback Generation Process.**   When invoked, the agent's `process` method coordinates the entire feedback generation workflow:

```python
def process(self, task_spec, verification_results, simulation_results,
            evaluation_results, model_plan, generated_code,
            current_code, previous_code, iteration,
            historical_fix_log):
```

The process follows these key steps:

- **Step 1: Generate Code Diff for Context**. If both previous and current code versions are available, the agent creates a diff to understand what has changed between iterations:

```python
        if previous_code and current_code:
            prev_lines = previous_code.splitlines(keepends=True)
            curr_lines = current_code.splitlines(keepends=True)

            diff = difflib.unified_diff(
                prev_lines, curr_lines,
                fromfile=f"simulation_code_iter_{iteration-1}.py",
                tofile=f"simulation_code_iter_{iteration}.py",
                n=3  # Context lines
            )
            code_diff = "".join(diff)
```

  This diff provides critical context for understanding how the code has evolved and whether previous feedback was incorporated.

- **Step 2: Update Historical Fix Log.** Before generating new feedback, the agent checks if previously identified issues have been fixed in the current code:

```
1    if iteration > 0 and historical_fix_log and current_code:
2        self._check_fixed_issues(historical_fix_log, current_code,
             iteration)
```

This check uses a specialized prompt that compares the current code against previously documented issues to determine if they've been resolved.

- **Step 3: Generate Comprehensive Feedback.** The agent uses the BaseAgent's `_build_prompt` method which fills a template with all the available information. The template (`feedback_generation_prompt.txt`, as shown in Prompt J.5) is structured to guide the LLM to perform a comprehensive analysis with the following workflow:

    1. **Code Analysis**: The LLM examines the current code and recent changes for:
        - Logical errors in the implementation
        - Specific syntactic or semantic errors
        - Execution flow issues
        - Appropriateness of data structures and algorithms
        - Issues with dependencies
    2. **Results Integration**: The feedback incorporates:
        - Verification results from code static analysis
        - Simulation execution results
        - Evaluation results comparing simulation output to expected behavior
    3. **Structured Output Generation**: The feedback is structured as a JSON object with:
        - `summary`: Brief overview of the feedback.
        - `critical_issues`: Issues that must be addressed.
        - `model_improvements`: Conceptual improvements to the simulation model.
        - `code_improvements`: Specific code modifications
        - `data_alignment_suggestions`: Ways to align the simulation with real-world data
        - `prioritized_actions`: Ordered list of recommended next steps.
        - `code_snippets`: Specific before/after code examples.

- **Step 4: Response Parsing and Validation.** After receiving the LLM's response, the agent processes it:

```
1    # Parse the LLM response and validate required schema fields
2    parsed = self._parse_llm_response(llm_response)
3    if not isinstance(parsed, dict) or "summary" not in parsed:
4        self.logger.warning("LLM feedback format is invalid, using
             placeholder feedback")
5        feedback = self._create_placeholder_feedback()
6    else:
7        feedback = parsed
```

If the LLM's response doesn't meet the expected format, a placeholder feedback structure is provided to ensure the workflow doesn't break.

**3. Historical Issue Tracking.** A unique feature of the Feedback Generation Agent is its historical issue tracking mechanism:

```
1    def _check_fixed_issues(self, historical_fix_log, current_code,
         current_iteration):
2        # Create prompt to check fixed issues
3        prompt = self._build_fix_check_prompt(historical_fix_log,
             current_code, current_iteration)
4
5        # Call LLM to analyze fixes
6        llm_response = self._call_llm(prompt)
7
```

```
8      # Parse response and update historical fix log
9      fixed_issues = self._parse_fix_check_response(llm_response)
10     if fixed_issues:
11         for iteration_key, issues in fixed_issues.items():
12             for fixed_issue in issues:
13                 if fixed_issue.get("status") == "fixed":
14                     # Update status in historical log
15                     # ...
```

This system:

- Maintains a record of issues across iterations.
- Uses the LLM to determine if each issue has been fixed in the current code.
- Updates the historical log with the status of each issue.
- Provides this context to future iterations to prevent re-introducing fixed issues.

The agent takes special care to handle the JSON parsing of LLM responses, with multiple fallback strategies to ensure robust operation even when responses are imperfectly formatted.

**4. Output Generation.** The final output is a comprehensive feedback structure that guides the Code Generation Agent in creating improved code for the next iteration. The feedback includes:

- Critical issues that need immediate attention.
- Model improvements for better simulation accuracy.
- Code improvements with specific snippets demonstrating the changes.
- Data alignment suggestions to better match real-world behavior.
- A prioritized action plan.

This structured feedback creates a clear roadmap for the next iteration, enabling continuous improvement of the simulation over time.

The **FGA**'s sophisticated approach to analyzing code, tracking issues over time, and providing structured, actionable feedback is central to SOCIA's ability to iteratively improve simulations through a multi-agent workflow.

# E THE DESIGN OF THE USER MODELING SIMULATOR

In this section, we provide illustration of how the structure, parameter, and calibration schema are designed in the **User Modeling** task.

**Black-box Problem.** In the **User Modeling** task, a key challenge arises from the simulator architecture: user ratings must be generated by an LLM treated as a **black-box component**. LLM module is callable but *non-differentiable* and lacks an *analytic likelihood*, making classical approaches such as gradient descent or maximum likelihood estimation inapplicable. In contrast, **Bayesian Optimization (BO)** and **Evolutionary Algorithms (EA)** can directly search for optimal parameter values on a predefined loss, providing point estimates. **SBI** offers an alternative by approximating the posterior distribution through a simulate–compare–learn paradigm, enabling parameter estimation and uncertainty quantification under black-box conditions. Consequently, in the **User Modeling** task, SBI, BO, and EA are viable choices, whereas gradient-based and likelihood-based methods are not.

**One-step predictor problem.** In this task, the simulator adopts an *implicit one-step predictor*, which means observations cannot be constructed as time-series rollouts. To solve the *one-step predictor* problem and design the appropriate observation vector, with the help of our CoS hints, SOCIA itself defines the observation vector as follows. Within SBI, we have:

$$\phi(x) = [\text{user history statistics} \parallel \text{aggregated scores of the target item by other users} \parallel \text{true rating } y],$$

with the goal of learning which parameter distributions are consistent with observed user/item features and true ratings. In contrast, for EA/BO calibration, the input features are:

$$\phi(x) = [\text{user history statistics} \parallel \text{aggregated scores of the target item by other users}],$$

and the objective is to find the parameter point that minimizes prediction error.

**How does SOCIA design a structure to handle black-box and one-step predictor challenges?** The simulator itself constructs a unified natural-language prompt from "user history + target item history" and feeds it to the LLM, which outputs a black-box score $s_{\text{llm}} \in [1, 5]$. This raw score is then mapped through a structured calibration head that both corrects systematic LLM biases and integrates structured features: (i) **Affine correction:** $\tilde{s} = a \cdot s_{\text{llm}} + b$, $\theta_{\text{aff}} = \{a, b\}$; (ii) **Local linear fusion:** $\hat{s} = \alpha \cdot \tilde{s} + (1 - \alpha) \cdot \sigma(w^\top \phi(x) + c) \times 4 + 1$, $\theta_{\text{lin}} = \{\alpha, w, c\}$; (iii) **Hierarchical bias (user/item bias):** $\hat{s} \leftarrow \hat{s} + u_{\text{user}} + v_{\text{item}}$, $\theta_{\text{bias}} = \{u_{\text{user}}, v_{\text{item}}\}$; (iv) **Observation noise:** $\epsilon \sim \mathcal{N}(0, \sigma^2)$, with parameter $\sigma$.

The full set of calibratable parameters is

$$\theta = \underbrace{\{a, b\}}_{\text{LLM correction}} \cup \underbrace{\{\alpha, w, c\}}_{\text{Feature fusion}} \cup \underbrace{\{u_{\text{user}}, v_{\text{item}}\}}_{\text{Hierarchical bias}} \cup \underbrace{\{\sigma\}}_{\text{Noise}}.$$

The final calibrated prediction is produced by: $\hat{s} = \text{CalibratorHead}(s_{\text{llm}}, \phi(x); \theta)$. This design preserves the LLM's strong semantic judgment capability while aligning its outputs with real ratings through interpretable and calibratable heads.

# F THE DESIGN OF THE MASK ADOPTION SIMULATOR

In this section, we provide illustration of how the structure, parameter, and calibration schema are designed in the **Mask Adoption** task.

**Mathematical Framework Overview** This simulation system models the spread of mask-wearing behavior through a social network using a combination of social influence, information propagation, and individual decision-making processes. The mathematical framework operates on multiple temporal and network scales.

**Core Mathematical Components**

**1. Agent State Representation Binary State Variable:**

- $s_i(t) \in \{0, 1\}$: Agent $i$'s mask-wearing state at time $t$

- $s_i(t) = 1$ if agent $i$ is wearing a mask at time $t$, $s_i(t) = 0$ otherwise

**Agent Attributes:**

- $r_i$: Risk perception of agent $i$ (individual characteristic)
- $\mathbf{d}_i$: Demographic vector (age group, occupation) encoded as one-hot vector

**2. Social Network Structure    Multilayer Network:**

- $G = \{G_F, G_W, G_C\}$: Three-layer network (Family, Work/School, Community)
- $N_i^{(l)}$: Set of neighbors of agent $i$ in layer $l \in \{F, W, C\}$
- $|N_i^{(l)}|$: Number of neighbors in layer $l$

**Neighbor Influence Computation:**

For each layer $l$ and time $t$, the social influence is computed as:

$$I_i^{(l)}(t) = \frac{1}{|N_i^{(l)}|} \sum_{j \in N_i^{(l)}} s_j(t)$$

This represents the fraction of neighbors in layer $l$ who are wearing masks at time $t$.

**3. Information Propagation Model    Information Reception:**

Agent $i$ receives information with probability:

$$P(\text{receive info}_i(t)) = 1 - \exp\left(-\sum_{l \in \{F,W,C\}} \phi_l \cdot I_i^{(l)}(t-1) - \lambda(t)\right)$$

Where:

- $\phi_l$: Information transmission rate for layer $l$
- $\lambda(t)$: Exogenous information broadcast rate at time $t$

**Memory Decay:**

Information memory follows exponential decay:

$$m_i(t) = \rho \cdot m_i(t-1) + (1 - \rho) \cdot \text{info}_i(t)$$

Where:

- $m_i(t)$: Memory of information at time $t$
- $\rho$: Memory decay parameter
- $\text{info}_i(t) \in \{0, 1\}$: Whether agent $i$ received information at time $t$

**4. Decision-Making Model    Logistic Decision Function:**

The probability that agent $i$ wears a mask at time $t$ is:

$$P(s_i(t) = 1) = \sigma\left(\text{logit}_i(t)\right)$$

Where $\sigma(x) = \frac{1}{1+e^{-x}}$ is the sigmoid function.

**Logit Computation:**

The logit for agent $i$ at time $t$ is:

$$\text{logit}_i(t) = \alpha + \gamma \cdot s_i(t-1) + \sum_{l \in \{F,W,C\}} \theta_l \cdot I_i^{(l)}(t-1) + \beta_r \cdot r_i + \beta_i \cdot m_i(t) + \mathbf{d}_i^T \mathbf{w}_d$$

Where:

- $\alpha$: Baseline adoption probability (intercept)
- $\gamma$: State persistence (inertia effect)
- $\theta_l$: Social influence weight for layer $l$
- $\beta_r$: Risk perception sensitivity
- $\beta_i$: Information influence strength
- $\mathbf{w}_d$: Demographic effect vector

### 5. Temporal Dynamics    State Transition:

The actual state is determined by:

$$s_i(t) = \begin{cases} 1 & \text{if random} < P(s_i(t) = 1) \\ 0 & \text{otherwise} \end{cases}$$

**Temperature Scaling (Optional):**

For stochastic behavior control:

$$\text{logit}_i(t) \leftarrow \frac{\text{logit}_i(t)}{\tau}$$

Where $\tau$ is the temperature parameter controlling decision randomness.

### 6. Government Intervention    Exogenous Signal:

The broadcast rate changes after intervention day $T_{\text{gov}}$:

$$\lambda(t) = \begin{cases} \lambda_{\text{base}} & \text{if } t < T_{\text{gov}} \\ \lambda_{\text{base}} \cdot f_{\text{gov}} & \text{if } t \geq T_{\text{gov}} \end{cases}$$

Where:

- $\lambda_{\text{base}}$: Base information broadcast rate
- $f_{\text{gov}}$: Government intervention amplification factor

**Mathematical Flow of the Simulation    Step 1: Information Propagation**

For each agent $i$ and time $t$:

1. Compute social influence: $I_i^{(l)}(t-1) = \frac{1}{|N_i^{(l)}|} \sum_{j \in N_i^{(l)}} s_j(t-1)$
2. Calculate information reception probability
3. Update memory: $m_i(t) = \rho \cdot m_i(t-1) + (1-\rho) \cdot \text{info}_i(t)$

**Step 2: Decision Making**

For each agent $i$ and time $t$:

1. Compute logit: $\text{logit}_i(t) = \alpha + \gamma \cdot s_i(t-1) + \sum_l \theta_l \cdot I_i^{(l)}(t-1) + \beta_r \cdot r_i + \beta_i \cdot m_i(t) + \mathbf{d}_i^T \mathbf{w}_d$
2. Apply temperature scaling if needed
3. Calculate probability: $P_i(t) = \sigma(\text{logit}_i(t))$
4. Sample state: $s_i(t) \sim \text{Bernoulli}(P_i(t))$

**Step 3: Aggregate Behavior**

The population-level adoption rate at time $t$ is:

$$R(t) = \frac{1}{N} \sum_{i=1}^{N} s_i(t)$$

Where $N$ is the total number of agents.

## Parameter Relationships and Effects  Social Influence Dynamics

- **High $\theta_l$ values**: Strong social influence from layer $l$
- **Network density effects**: Denser networks amplify social influence
- **Layer interaction**: Multiple layers create compound effects

### Individual Characteristics

- **Risk perception** ($\beta_r$): Higher values increase mask adoption for risk-averse agents
- **Demographic effects** ($\mathbf{w}_d$): Age and occupation create heterogeneous adoption patterns
- **Memory effects** ($\beta_i, \rho$): Information persistence affects long-term behavior

### Temporal Dynamics

- **Inertia** ($\gamma$): High values create state persistence
- **Memory decay** ($\rho$): Controls how long information affects decisions
- **Intervention timing**: $T_{\text{gov}}$ and $f_{\text{gov}}$ create temporal discontinuities

## Calibration and Optimization  Parameter Estimation

The system estimates parameters $\Theta = \{\alpha, \gamma, \theta_F, \theta_W, \theta_C, \beta_r, \beta_i, \mathbf{w}_d, \phi_F, \phi_W, \phi_C, \lambda_{\text{base}}, f_{\text{gov}}, \rho, \tau\}$ by minimizing:

$$\mathcal{L}(\Theta) = \sum_{t=1}^{T} \left(R_{\text{observed}}(t) - R_{\text{simulated}}(t, \Theta)\right)^2 + \lambda_{\text{reg}} \|\Theta\|_2^2$$

### Multi-Objective Optimization

The calibration process optimizes multiple metrics simultaneously:

- **RMSE**: Root mean square error of adoption rates
- **MAE**: Mean absolute error of temporal trajectories
- **Brier Score**: Probability calibration quality
- **Transition Fit**: Accuracy of state transition probabilities

## Mathematical Properties  Convergence Properties

- **Memory decay**: $\lim_{t \to \infty} m_i(t) = \frac{1-\rho}{\rho} \cdot \mathbb{E}[\text{info}_i(t)]$ (if stationary)
- **Social influence**: Bounded by network structure and parameter values
- **State transitions**: Follow Markovian properties with social memory

### Scale Effects

- **Network size**: Effects scale with $\frac{1}{|N_i^{(l)}|}$ for each layer
- **Population size**: Aggregate behavior converges to expected values
- **Temporal scale**: Discrete time steps approximate continuous dynamics

This mathematical framework provides a comprehensive foundation for modeling social behavior propagation through networks, with explicit connections between individual characteristics, social influence, information dynamics, and population-level outcomes.

# G  THE DESIGN OF THE PERSONAL MOBILITY SIMULATOR

In this section, we provide illustration of how the structure, parameter, and calibration schema are designed in the **Personal Mobility Generation** task.

**1) Data → features**

- **Trajectories**: resident-day strings (e.g., "Rest Area#1120 at 10:50:00 → . . . "), parsed into ordered steps $(l_i, t_i)$, where $l_i$ carries a POI **ID**, **name**, and later (**lat, lon**); **time** is wall-clock within day.
- **Categories**: map POI base names (e.g., "Rest Area") → activity categories (e.g., "Travel & Transport") for semantic routines.
- **Coordinates**: from your POI→(lat,lon,full name) mapping to enable distance/time and spatial histograms.

**2) Simulator structure (agent-based, two-stage choice + dwell) Time resolution:** discrete 10-minute ticks ($T = 144$ per day; adjustable).
**Agent state $s_t$:** current location $l_t$, category $c_t$, clock $t$, day-type (weekday/weekend), **regime** $r \in \{\text{normal}, \text{pandemic}\}$, personal prefs $\pi$, mobility budget $B$, memory $m_t$ (recent visits), compliance $q$.

**2.1 Activity-type choice (what to do next?)** A multinomial logit over categories $c$:

$$P(c \mid s_t) \propto \exp \Big( \theta_0^{(c)} + \theta_{\text{time}}^{(c)} \, \phi_{\text{time}}(t) + \theta_{\text{dow}}^{(c)} \, \phi_{\text{dow}}$$
$$+ \theta_{\text{home}}^{(c)} \, \mathbb{1}[c = \text{Residence}] + \xi \, \text{RepeatBias}(c, m_t)$$
$$+ \delta \, \text{WeekendShift} + \kappa_r^{(c)} \Big)$$

- $\kappa_r^{(c)}$ is a **regime modifier** (closures/avoidance during pandemic).

**2.2 Destination choice (where to do it?)** Conditional on $c$, pick a specific POI $l \in \text{POIs}(c)$ via a gravity/saliency model:

$$P(l \mid c, s_t) \propto \exp \Big( \alpha_c \, \text{POIPop}(l) - \beta_c \, \text{dist}(l_t, l) + \eta_c \, \text{HomeAnchor}(l) + \kappa_r^{(l)} \Big)$$

- dist uses Haversine between POI coordinates; POIPop can be proxied by frequency in data.
- $\kappa_r^{(l)}$ applies POI-level closures/capacity limits.

**2.3 Dwell & move**

- **Dwell time** at POI $l$: LogNormal($\mu_c, \sigma_c$) (category-specific), capped by budget $B$.
- **Travel time** between $l_t \to l$: dist/$v$ with regime-dependent speed $v_r$ (congestion/curfew effects).
- Advance clock by travel+dwell; append step $(l, t)$.

**3) Regime & policy layer (to enable WHAT-IF)** Represent pandemic effects with interpretable **structural multipliers**:

- **Category availability/avoidance**: $\kappa_{\text{pandemic}}^{(c)} \in [-\infty, 0]$ (closures/aversion)
- **Mobility reduction**: $\rho_{\text{mob}} \in (0, 1]$ scales $B$ and increases home bias $\eta_{\text{home}}$
- **Curfew**: forbidden windows $[t_{\min}, t_{\max}]$ (probabilistic compliance $q$)
- **Remote-work prob.** $p_{\text{remote}}$ (pushes work-category demand to "Home (private)")
- **Contact/venue crowding cap**: down-weights POIPop in high-risk categories (e.g., nightlife).

**Scenarios**

1. **Normal→Normal**: set $r = $ normal, calibrate $\theta, \alpha, \beta, \mu, \sigma, \dots$ on 2019.

2. **Pandemic→Pandemic**: $r = $ pandemic, learn modifiers $\kappa, \rho, q, p_{\text{remote}}, \dots$ on 2021.

3. **Normal→Pandemic (WHAT-IF)**: train base on 2019; at test time **switch on** pandemic multipliers with priors learned from limited 2021 aggregates (or policy inputs). Structural separation keeps extrapolation stable.

**4) Calibratable parameter set (works for SBI / BO / EA) Global**

$$\text{dt\_minutes}, \quad v_{\text{normal}}, \quad v_{\text{pandemic}}, \quad \text{budget } B$$

**Activity-type**

$$\theta_0^{(c)}, \quad \theta_{\text{time}}^{(c)}, \quad \theta_{\text{dow}}^{(c)}, \quad \theta_{\text{home}}, \quad \text{repeat\_bias } \xi, \quad \text{weekend\_shift } \delta$$

**Destination**

$$\alpha_{\text{pop}}^{(c)}, \quad \beta_{\text{dist}}^{(c)}, \quad \eta_{\text{home}}$$

**Dwell**

$$\mu^{(c)}, \quad \sigma^{(c)}$$

**Regime modifiers**

$$\kappa_{\text{cat}}^{(c)} \ (\leq 0 \text{ in pandemic}), \quad \kappa_{\text{poi}}^{(l)}, \quad \rho_{\text{mobility}} \in (0, 1], \quad \text{curfew } (t_{\text{start}}, t_{\text{end}}, q), \quad p_{\text{remote}}$$

**Stochasticity**

$$\epsilon_{\text{explore}}, \quad \gamma_{\text{memory\_decay}}$$

**Priors (for SBI/BO/EA)**: weakly informative (e.g., $\beta_{\text{dist}} > 0$, $\sigma_c > 0$; $\kappa \leq 0$ for restricted cats; $\rho \in (0, 1]$).

**Vectorization for SBI**: convert per-day outputs into fixed-length vectors (histograms and summary features in §5) so neural posterior estimators can ingest them.

**5) Calibration loop**

1. **Sample** resident $u$ and date $d$ from 2019 (normal) or 2021 (pandemic).

2. **Simulate** a day with regime $r$ (switch $r$ to pandemic for what-if).

3. **Summaries**: compute SD, SI, DARD $(t, c)$ histogram, STVD $(t, \text{lat}, \text{lon})$ histogram, plus optional add-ons.

4. **Discrepancy**: weighted sum of divergences (e.g., JSD for histograms, DTW for paths).

5. **Optimizer**:
   - **SBI**: learn $p(\theta \mid s)$ using the vectorized summaries.
   - **Bayesian Opt.**: minimize discrepancy; handle 20–50 dims with GP-EI/TPE.
   - **EA**: evolve parameter vectors; keep hard constraints (e.g., $\beta > 0$).

**6) Train/Val/Test splits aligned to scenarios**

- **Normal model**: train on 2019 resident-days; validate on held-out 2019; test on 2019.

- **Pandemic model**: train/val/test on disjoint 2021 days.

- **What-if**: train on 2019; validate by **policy-switched replays** on a small 2019 subset (synthetic restrictions); final evaluate on **2021** days (no parameter re-fitting; only switch regime multipliers).

**7) Practical notes**

- Parse base names by stripping the "#ID" suffix to join with categories (e.g., "Rest Area#1120" → "Rest Area" → "Travel & Transport").

- Identify **Home** anchors using category "Home (private)" (POIs like "Home#736" appear in 2021 examples), which is crucial for home bias and nightly return.

**Verdict on four metrics**

- Use **JSD** for DARD/STVD histograms, **mean/variance** for SD/SI, and add **DTW / edit distance + entropy/gyration** as secondary diagnostics. This set is expressive enough for SBI and stable for BO/EA.

# H AN EXAMPLE OF BLUEPRINT

In this section, we provide an illustrative example of a complete blueprint generated by the **Data Analysis Agent** for the **Mask Adoption Behavior** scenario. This structured JSON object contains all the necessary information, from data analysis summaries to simulation design parameters, required by downstream agents to construct and execute the simulator.

Generated Blueprint: Mask Adoption Behavior

```json
{
  "title": "Simulation Task",
  "description": "Develop a multi-agent simulation system that
      models the spread of mask-wearing behavior through social
      networks.",
  "simulation_focus": [
    "Predict agents' mask-wearing behavior over day 30 - day 39 (
        last ten days) based on their attributes, social connections
        , and past behavior",
    "Model information propagation and behavioral changes triggered
        by social influence and government intervention starting on
        Day 10"
  ],
  "data_folder": "data_fitting/mask_adoption_data/",
  "data_files": {
    "agent_attributes.csv": "Contains demographic and behavioral
        attributes of each agent, including age, occupation, risk
        perception, and social connection counts",
    "social_network.json": "Contains structured data representing a
        social network. Each entry corresponds to a user (identified
         by a unique integer ID), and the values describe the
        different types of relationships that user has with others.
        Data Structure: The top-level structure is a dictionary ({}
        in JSON). Keys: Each key is a unique user ID (as an integer,
         e.g., 0, 1, 2, etc.). Values: Each value is another
        dictionary with the following fields: \"family\": a list of
        user IDs representing family connections. \"work_school\": a
         list of user IDs representing colleagues or schoolmates. \"
        community\": a list of user IDs representing community-level
         acquaintances. \"all\": a merged list of all user IDs
        connected to the user across the three categories above.",
    "train_data.csv": "Time series data for the first 30 days, used
        for training the model"
  },
  "evaluation_metrics": {
    "RMSE": {
      "description": "Root Mean Square Error measuring overall
          predictive accuracy of mask adoption rates",
      "interpretation": "Directly interpretable as percentage
          deviations due to the 0-1 range",
      "formula": "RMSE = sqrt(sum((predicted_rate - actual_rate)^2)
          / n)"
    }
  },
  "data_analysis_result": {
    "overall_simulation_design": {
      "objective": "Model and forecast the spread and persistence of
           mask-wearing behavior across a multilayer social network
          via social influence, individual risk perception, and
          exogenous information signals; automatically calibrate
          behavioral parameters to longitudinal observations.",
      "initialization": "1) Load agents from agent_attributes.csv (
          IDs, demographics, risk_perception, initial_mask_wearing).
           2) Load network from social_network.json; build multiplex
```

```
                    layers (family, work_school, community), symmetrize and
                    deduplicate edges. 3) Align populations across files; use
                    intersection of IDs. 4) Set day 0 mask state per agent
                    from train_data.csv when available; otherwise fallback to
                    initial_mask_wearing. 5) Derive initial received_info from
                     train_data day 0 if provided; else initialize false. 6)
                    Precompute per-agent layer degrees and normalization
                    constants. 7) Initialize parameter vector (layer weights,
                    inertia, risk sensitivity, info diffusion, exogenous
                    broadcast rate, decay) with priors or heuristic defaults
                    .",
25          "execution": "Daily synchronous updates. At each time step,
                    compute per-agent neighbor mask share per layer; generate
                    received_info via peer and exogenous channels; compute
                    mask adoption probability using a calibrated logistic
                    decision rule with inertia and demographics. Calibrate
                    parameters by maximizing the likelihood of observed
                    wearing_mask transitions on the training time window (
                    penalized logistic regression over panel data) while
                    matching observed received_info rates; use gradient-based
                    optimization or Bayesian optimization with cross-validated
                     early stopping. After calibration, run forward
                    simulations on the validation window from the last
                    observed state, then evaluate.",
26          "outputs": [
27            "Calibrated parameter set (point estimates and, if enabled,
                    posterior samples/uncertainty)",
28            "Validation forecasts of per-agent and aggregate mask-
                    wearing trajectories",
29            "Evaluation metrics on validation set (RMSE, MAE, Brier,
                    TransitionFit)",
30            "Calibration diagnostics (convergence stats, parameter
                    sensitivities, regularization paths)",
31            "Reproducible seeds and configuration used for this run"
32          ]
33        },
34        "scale_granularity": {
35          "time_step": "days",
36          "spatial_resolution": "non-spatial",
37          "population_size": "Number of unique agent_id present in both
                    agent_attributes.csv and social_network.json (expected
                    ~1000 based on samples)",
38          "rationale": "train_data.csv is daily; network interactions
                    are not spatially explicit; population equals provided
                    social graph."
39        },
40        "agent_archetypes": {
41          "unit": "resident",
42          "roles": [
43            "resident",
44            "campaign_broadcaster"
45          ],
46          "static_attributes": [
47            "agent_id",
48            "age",
49            "age_group",
50            "occupation",
51            "risk_perception",
52            "layer_degrees: family_connections, work_school_connections,
                    community_connections",
53            "demographic multipliers (derived, not raw fields)"
54          ],
```

```
55        "dynamic_states": [
56          "wearing_mask_t (bool)",
57          "received_info_t (bool)",
58          "memory_of_info_t (continuous 0-1, optional for decay)",
59          "propensity_score_t (latent probability from decision rule)"
60        ],
61        "construction_from_data": "Map each CSV row in
              agent_attributes.csv to an agent. Set initial
              wearing_mask_t from train_data day 0 if available, else
              from initial_mask_wearing. Risk perception is the provided
               [0,1] value. Age group and occupation are categorical
              features used as fixed effects or multipliers. Degrees per
               layer come from agent_attributes.csv for validation;
              actual edges come from social_network.json.",
62        "update_from_data": "During calibration, use train_data.csv to
               compute observed transitions (wear_t-1 -> wear_t) and
              received_info_t targets per day-agent. These observations
              fit the logistic decision model to estimate parameters.
              During simulation, dynamic states update from the model
              using simulated signals (neighbor influence and exogenous
              info) and the last state."
63      },
64      "interaction_topology": {
65        "topology": "graph",
66        "layers": [
67          "family",
68          "work_school",
69          "community",
70          "all (union, optional projection)"
71        ],
72        "protocol": "peer-to-peer interactions on each layer with
              layer-specific weights; synchronous daily update of states
              ",
73        "construction_from_data": "Use social_network.json adjacency
              lists per agent per layer. Symmetrize edges (if i lists j,
               ensure j lists i) and remove duplicates. Validate node
              IDs against agent_attributes.csv. If social_network.json
              is unavailable for some IDs, optionally synthesize edges
              using the degree targets in agent_attributes.csv via a
              configuration-model per layer respecting plausible
              homophily (e.g., students in work/school)."
74      },
75      "information_propagation": {
76        "exists": true,
77        "topology": "Multiplex diffusion across family, work_school,
              community; plus global broadcast channel",
78        "mechanism": "Each day, an agent receives information with
              probability: 1 - exp(-(phi_family * share_family -
              weighted by mask-wearing neighbors - + phi_work *
              share_work + phi_community * share_community +
              lambda_broadcast)), capped at [0,1]. Optionally maintain a
               decaying memory_of_info with rate rho_info_decay that
              sustains influence for several days.",
79        "construction_from_data": "If train_data.csv includes
              received_info, match the empirical daily and agent-level
              prevalence to calibrate lambda_broadcast and phi_* via
              likelihood or moment matching. If received_info is missing
               or sparse, infer via the peer mechanism only and
              calibrate to reproduce mask adoption dynamics.",
80        "parameters": {
81          "names": [
82            "phi_family",
```

```
 83            "phi_work",
 84            "phi_community",
 85            "lambda_broadcast",
 86            "rho_info_decay"
 87          ],
 88          "notes": "phi_* scale peer info exposure per layer;
                 lambda_broadcast is a daily rate for exogenous campaigns
                 ; rho_info_decay controls persistence of received_info.
                 Estimate from observed received_info (if present) and
                 mask transitions via joint likelihood."
 89        }
 90      },
 91      "exogenous_signals": [
 92        {
 93          "name": "received_info_broadcast",
 94          "construction_from_data": "train_data.csv received_info
                 field and its daily prevalence; if absent, treat as
                 latent and calibrate lambda_broadcast to fit adoption
                 curves.",
 95          "observability": "Residents can observe their own exposure;
                 no delay assumed; can include binomial noise.",
 96          "effect_on_agents": "Adds an exogenous term to adoption
                 probability and reduces abandonment probability (enters
                 decision rule as a positive coefficient).",
 97          "bounds": "[0,1] daily probability"
 98        }
 99      ],
100      "action_decision_policy": {
101        "by_role": {
102          "resident": {
103            "inputs": [
104              "previous wearing_mask state",
105              "neighbor mask share per layer (family, work_school,
                     community)",
106              "risk_perception",
107              "received_info (current or decayed memory)",
108              "demographic indicators: age_group, occupation",
109              "degree-normalized influence (optional)"
110            ],
111            "policy_form": "Logistic adoption-retention model. Let x_t
                   be a feature vector including: intercept; inertia I(
                   wear_t-1); neighbor shares per layer weighted by
                   w_family, w_work, w_community; risk_perception;
                   received_info (or memory); demographic fixed effects;
                   interaction terms (e.g., risk_perception *
                   received_info). Adoption probability P(wear_t=1 | x_t)
                   = sigmoid(alpha + gamma*I(wear_t-1) + beta_f*w_family
                   *share_family + beta_w*w_work*share_work + beta_c*
                   w_community*share_community + beta_r*risk + beta_i*
                   received_info + demographic terms). State is sampled
                   Bernoulli from this probability (or use deterministic
                   threshold for expectation runs).",
112            "parameters": [
113              "alpha (intercept)",
114              "gamma (inertia/persistence)",
115              "w_family, w_work, w_community (layer weights, sum-to-
                     one constraint applied during calibration)",
116              "beta_f, beta_w, beta_c (peer influence slopes per layer
                     )",
117              "beta_r (risk perception coefficient)",
118              "beta_i (info exposure coefficient)",
```

```
119              "age_group effects: beta_age_youth, beta_age_young_adult
                    , beta_age_middle_age, beta_age_senior (one is
                    baseline)",
120              "occupation effects: beta_occ_student,
                    beta_occ_blue_collar, beta_occ_white_collar (one
                    baseline)",
121              "tau (temperature/noise scaling, optional)"
122            ]
123          },
124          "campaign_broadcaster": {
125            "inputs": [
126              "target coverage or budget (optional)",
127              "current adoption rate (optional for feedback control)"
128            ],
129            "policy_form": "Stateless stochastic broadcast: each agent
                    independently receives info with probability
                    lambda_broadcast per day; optionally modulate
                    lambda_broadcast by time or adoption level.",
130            "parameters": [
131              "lambda_broadcast"
132            ]
133          }
134        }
135      },
136      "holdout_plan": {
137        "method": "temporal_holdout",
138        "train_range": "First 80% of days per agent in train_data.csv
                (e.g., if days 0-99, train on 0-79).",
139        "validation_range": "Last 20% of days per agent (e.g., days
                80-99).",
140        "notes": "Initialize validation simulation from the last
                observed train-day state per agent. Ensure no leakage of
                validation labels into calibration of parameters."
141      },
142      "simulation_evaluation": {
143        "metrics": [
144          {
145            "name": "RMSE_aggregate",
146            "definition": "Root Mean Squared Error between simulated
                    and observed daily population-level mask-wearing rates
                     on validation days."
147          },
148          {
149            "name": "MAE_aggregate",
150            "definition": "Mean Absolute Error between simulated and
                    observed daily adoption rates on validation."
151          },
152          {
153            "name": "Brier",
154            "definition": "Mean squared error between predicted per-
                    agent probabilities and observed wearing_mask outcomes
                     on validation."
155          },
156          {
157            "name": "TransitionFit",
158            "definition": "Absolute or squared error between simulated
                     and observed transition probabilities P01, P11, P10,
                    P00 on validation."
159          }
160        ],
161        "comparison_method": "Run K stochastic simulations on the
                validation window (e.g., K=50) and average metrics; report
```

```
                     mean+/-CI. Also report calibration plots: simulated vs
                     observed aggregate curve, and confusion/transition
                     matrices. Select best run by mean Brier or a weighted
                     composite of metrics."
162        },
163        "calibratable_parameters": [
164            {
165                "name": "w_family, w_work, w_community",
166                "range_bounds": "[0, 2] each; normalized to sum to 1 at
                     runtime",
167                "source": "Layer salience; constrained by network layers and
                      observed dynamics",
168                "notes": "Higher family weight typically increases
                     clustering-driven adoption."
169            },
170            {
171                "name": "beta_f, beta_w, beta_c",
172                "range_bounds": "[0, 5]",
173                "source": "Peer influence slopes per layer from train_data
                     transitions",
174                "notes": "Control sensitivity to neighbor mask share."
175            },
176            {
177                "name": "gamma (inertia)",
178                "range_bounds": "[0, 6]",
179                "source": "Persistence of behavior",
180                "notes": "Higher gamma reduces switching; calibrated from
                     P11 and P00."
181            },
182            {
183                "name": "beta_r (risk perception coefficient)",
184                "range_bounds": "[0, 5]",
185                "source": "agent_attributes.risk_perception",
186                "notes": "Scales individual propensity to wear independent
                     of peers."
187            },
188            {
189                "name": "beta_i (info exposure effect)",
190                "range_bounds": "[0, 5]",
191                "source": "train_data.received_info",
192                "notes": "Amplifies adoption when exposed to information."
193            },
194            {
195                "name": "lambda_broadcast",
196                "range_bounds": "[0, 0.5] daily",
197                "source": "Exogenous info prevalence in train_data",
198                "notes": "Baseline probability of receiving info independent
                      of peers."
199            },
200            {
201                "name": "phi_family, phi_work, phi_community",
202                "range_bounds": "[0, 2]",
203                "source": "Information propagation via peers",
204                "notes": "Map neighbor mask share to info receipt
                     probability."
205            },
206            {
207                "name": "rho_info_decay",
208                "range_bounds": "[0, 1]",
209                "source": "Duration of info effect",
210                "notes": "Per-day retention of received_info memory."
211            },
```

```
212        {
213          "name": "alpha (intercept)",
214          "range_bounds": "[-5, 5]",
215          "source": "Baseline adoption propensity",
216          "notes": "Centers probabilities absent other signals."
217        },
218        {
219          "name": "age_group effects (beta_age_*)",
220          "range_bounds": "[-3, 3]",
221          "source": "agent_attributes.age_group",
222          "notes": "Additive fixed effects; one category is baseline
                       at 0."
223        },
224        {
225          "name": "occupation effects (beta_occ_*)",
226          "range_bounds": "[-3, 3]",
227          "source": "agent_attributes.occupation",
228          "notes": "Additive fixed effects; one category baseline."
229        },
230        {
231          "name": "tau (temperature/noise)",
232          "range_bounds": "[0.5, 5]",
233          "source": "Controls stochasticity in decision sampling",
234          "notes": "Lower tau yields more deterministic behavior."
235        },
236        {
237          "name": "regularization (L2)",
238          "range_bounds": "[0, 10]",
239          "source": "Calibration stabilization",
240          "notes": "Prevents overfitting; tuned by inner CV on train
                       window."
241        }
242      ]
243    }
244  }
```

# I  AN EXAMPLE OF FEEDBACK FOR SIMULATOR OPTIMIZATION

In this section, we present an example of feedback generated during the second iteration of the **Mask Adoption Behavior** simulation. This feedback was produced by the **Feedback Generation Agent** based on the code generated in the previous iteration, and serves to guide subsequent improvements in the simulator.

> **Feedback Example: Mask Adoption Behavior - iteration 2**
>
> **summary:**
> The simulation code needs critical fixes to address execution errors and logical issues related to the influence propagation. The model should better reflect real-world interactions for improved simulations.
>
> **Critical issues:**
>
> - **issue:**  Incorrect operator usage for boolean values in influence propagation.
> - **impact:**  This causes execution failure due to unsupported operations, preventing any simulation results.
> - **solution:**  Replace the subtraction operator with a suitable boolean operation such as XOR for the mask-wearing status comparison.
> - **introduced_by_changes:**  false

**Model improvements:**

- **aspect:** Influence propagation model.
- **current_approach:** Uses a simple linear influence addition without considering varying influence strength based on relationship types.
- **suggested_approach:** Incorporate different influence weights based on relationship types (e.g., family, work/school, community) to better simulate real-world dynamics.
- **expected_benefit:** More accurate modeling of social influence, leading to better alignment with observed mask-wearing behaviors.

**Code improvements:**

1. Improvement
   - **file:** simulation.py
   - **modification:** Replace boolean subtraction with XOR in influence propagation.
   - **reason:** Corrects the execution error and enables the model to function as intended.
   - **related_to_recent_changes:** false
2. Improvement
   - **file:** simulation.py
   - **modification:** Replace dynamic decay and influence propagation logic to consider connection type weights.
   - **reason:** Enhances realism of social influence and behavioral change simulation.
   - **related_to_recent_changes:** false

**Data Alignment Suggestions:**

- **metric:** Adoption rate.
- **current_gap:** The model does not account for varying influence by connection type.
- **suggestion:** Introduce weights for different social connections to more accurately simulate influence dynamics.

**Prioritized Actions:**

- Fix boolean operation error in influence propagation.
- Introduce variable influence weights for different connection types.
- Refactor dynamic decay calculation to ensure logical consistency.

# J    PROMPTS IN SOCIA

In this section, we present the prompts used to trigger and guide each agent in SOCIA to perform specific functions or generate targeted outputs.

## J.1    CHAIN-OF-STRUCTURE PROMPT

**Chain-Of-Structure Prompt**

```
You are an expert data scientist and simulation modeler.  Your
task is to analyze data for designing a simulation model.  Overall
simulation design is:  automatically generate and calibrate the
simulator.  This requires defining the classes and functions within
the simulator, as well as how these classes and functions interact.
Next, define the parameters used by these functions and classes.  The
inputs are the data together with the initialization of parameters.
The execution involves tuning parameters through data calibration.
The final outputs are the calibrated parameters and the evaluation
results on the validation dataset.
```

**TASK DESCRIPTION:**
{task_description}

**DATA SUMMARIES:**
{file_summaries_text}

```
Based on the provided task description and data summaries, please
analyze the data and provide guidelines for simulation model
construction.

Your analysis must cover:

    1. Overall simulation design:  State the primary objective of
       the simulation.  Specify how the simulation initializes from
       inputs.  Specify the execution of the simulation (should
       involve tuning parameters through data calibration).  Specify
       what artifacts it outputs after execution (should be the
       calibrated parameters and the evaluation results on the
       validation dataset).

    2. Scale & Granularity:  Specify time step, spatial resolution (or
       explicitly "non-spatial"), and population size (agents), with
       brief rationale.

    3. Agent Archetypes:  Define the agent unit and roles; list static
       attributes and dynamic states; explain how the input data
       construct static attributes and how dynamic states update from
       data-derived signals.

    4. Interaction Topology:  Describe how agents interact; explain
       how to build interactions (layers/edges/protocols) from the
       input data.

    5. Information propagation:  Clarify whether information diffusion
       exists, its topology/mechanism, and how to parameterize/drive
       it using inputs.

    6. Exogenous Signals:  Identify any external
       signals/interventions, how they are derived from inputs, and
       how they affect agent decisions.

    7. Action Decision Policy:  Describe actions taken by agents and
       the decision policy mapping observations/signals to actions;
       include role-specific inputs and policy forms.

    8. Holdout:  Propose a training / validation data split plan if
       needed.  For time series, prefer first 80% of days as train
       and last 20% as validation; state exact ranges or the rule to
       compute them.
```

9. **Simulation Evaluation**: Define evaluation metrics and how to compare simulator output artifacts against validation ground truth.

10. **Calibratable Parameters**: List tunable parameters with bounds/ranges that will be used for configuring the simulation.

Provide your response in the following JSON format (valid JSON only, no extra text):

```
{
  "overall_simulation_design": {
    "objective": "what is this simulation about",
    "initialization": "How the simulation starts from inputs (states,
        seeds, signals)",
    "execution": "How to tune parameters through data calibration",
    "outputs": ["List of simulation artifacts to produce (e.g., the
        calibrated parameters and the evaluation results on the
        validation dataset)"]
  },
  "scale_granularity": {
    "time_step": "seconds|minutes|hours|days",
    "spatial_resolution": "non-spatial|grid|POI|road network|other
        :<...>",
    "population_size": "integer or description tied to data",
    "rationale": "Why these scales are appropriate"
  },
  "agent_archetypes": {
    "unit": "e.g., resident",
    "roles": ["role1", "role2"],
    "static_attributes": ["list of static fields mapped from inputs
        "],
    "dynamic_states": ["list of dynamic states tracked over time"],
    "construction_from_data": "How input data map to static
        attributes (IDs, demographics, risk, initial states, etc.)",
    "update_from_data": "How dynamic states update using observations
        /signals derived from data"
  },
  "interaction_topology": {
    "topology": "graph|hybrid|broadcast|market|spatial",
    "layers": ["e.g., family", "work_school", "community", "all"],
    "protocol": "p2p on layers|group broadcast|platform-level rules",
    "construction_from_data": "How to build edges/layers from
        social_network / counts / metadata"
  },
  "information_propagation": {
    "exists": true,
    "topology": "layer(s) used for diffusion",
    "mechanism": "e.g., neighbor fraction, threshold, broadcast rate
        ",
    "construction_from_data": "Which file/field drives info intensity
        or schedule",
    "parameters": {"names": ["e.g., info_rate", "neighbor_weight"], "
        notes": "how to estimate or tune"}
  },
  "exogenous_signals": [
    {
      "name": "signal_name",
      "construction_from_data": "which file/field defines or
          constrains it",
      "observability": "which roles can observe; delay/noise if any",
      "effect_on_agents": "how it enters decision function",
      "bounds": "[low, high]"
    }
```

```
43        ],
44      "action_decision_policy": {
45        "by_role": {
46          "role1": {
47            "inputs": ["observations/signals used by this role"],
48            "policy_form": "e.g., logistic with peer/self/policy terms;
                 thresholds; inertia",
49            "parameters": ["list of per-role parameters"]
50          },
51          "role2": {
52            "inputs": ["..."],
53            "policy_form": "...",
54            "parameters": ["..."]
55          }
56        }
57      },
58      "holdout_plan": {
59        "method": "temporal_holdout|rolling_backtest",
60        "train_range": "e.g., day 0-23 or first 80% rule",
61        "validation_range": "e.g., day 24-29 or last 20% rule",
62        "notes": "any agent-level holdout or rolling details"
63      },
64      "simulation_evaluation": {
65        "metrics": [
66          {"name": "RMSE_aggregate", "definition": "RMSE between
                 simulated and observed daily adoption rates"},
67          {"name": "MAE_aggregate", "definition": "MAE between curves"},
68          {"name": "Brier", "definition": "Brier score for per-agent
                 probabilities if available"},
69          {"name": "TransitionFit", "definition": "error on P10/P11/P01
                 rates vs observed"}
70        ],
71        "comparison_method": "how to compute metrics on validation set
                and report (tables/plots)"
72      },
73      "calibratable_parameters": [
74        {
75          "name": "influence_weight_family|work_school|community",
76          "range_bounds": "[0,1] or rationale",
77          "source": "constrained by network layers / train_data dynamics
                 ",
78          "notes": "tie to topology and adoption speed"
79        },
80        {
81          "name": "threshold|inertia|policy_weight|info_rate|memory_decay
                 ",
82          "range_bounds": "[0,1] or problem-appropriate bounds",
83          "source": "estimated from transitions; refined via optimization
                 ",
84          "notes": "regularize to avoid overfitting"
85        }
86      ]
87    }

  Return only valid JSON that can be parsed.  Do not include any other
  explanation or text outside the JSON.
```

**Prompt:** *This is the prompt for the **Chain of Structure**.*

## J.2  CODE GENERATION PROMPT

---

### Code Generation Prompt

You are the Code Generation Agent in a system that generates social simulations.  Transform the provided Blueprint into production-grade, executable Python code.

Please read the following instructions for the code generation task.

0. **Global Requirements:**
    - Write clean, modular, PEP-8 compliant code with complete docstrings (triple-quoted, not truncated).
    - Provide full class/function bodies (no stubs).
    - The output must be a single standalone Python program that runs end-to-end without manual edits.
    - Ensure deterministic behavior via a global random seed.
    - Validate all inputs; raise clear exceptions with actionable messages.

1. **Path Handling Instructions**

    When generating code, setup data file paths as follows:

```python
import os
PROJECT_ROOT = os.environ.get("PROJECT_ROOT")
DATA_PATH = os.environ.get("DATA_PATH")
DATA_DIR = os.path.join(PROJECT_ROOT, DATA_PATH)
```

    For ALL data files mentioned in the Task Specification, use this consistent path format:

```python
agent_file = os.path.join(DATA_DIR, "agent_attributes.csv")
```

2. **Orchestrator (Non-Empty Main Flow):**
    - The generated code must include a non-empty main() function that acts as the orchestrator.
    - main() must call, in order:  parse_cli() (which is optional) -> load_data() -> build_network_and_agents() -> holdout_split() -> calibrator.fit() -> simulator.rollout() -> evaluator.compute_metrics() -> save_results().
    - main() must not be empty, must not contain only pass, and must allow the program to be executed end-to-end directly with python simulate.py ....

    Then, as an expert software engineer and simulation modeler, you need to generate production-grade Python code for a multi-agent simulator that follows the Blueprint below.  You should implement data ingestion, agent modeling, multilayer interactions, information propagation, exogenous signals, temporal holdout, a pluggable calibration algorithm, forward simulation on the validation window, and metric evaluation.

    Transform the Blueprint below into production-grade executable Python code.

    **Blueprint is:**
    {blue_print}

    When generating the code following the Blueprint, please also achieve the following requirements.

    According to the blueprint's guidance, we need to:
    - Read the input data files from absolute paths.

---

- Following the blueprint's guidance, construct the simulator code. Based on the data files, model the various elements of the simulator as designed in the blueprint, and generate the outputs specified by the blueprint.
- According to the blueprint's guidance, partition the training data (Holdout).
- Develop a calibration algorithm aimed at fitting (calibrating) the simulator's parameters. The evaluation interface should remain stable, and the calibration algorithm should be modular and replaceable.
- Using the training portion of the data, apply the calibration algorithm to estimate/fit the simulator's parameters, so that the simulator aligns as closely as possible with the real data on this segment.
- Using the validation portion of the data, allow the model to forward simulate (roll out). Compare the simulation results against the real data according to the evaluation metrics defined in the blueprint, and compute the error values.

3. **Pluggable Calibration Architecture (MANDATORY)**
   - Implement the following template and contracts verbatim (package-local, in the same file):

   Template:

```python
from abc import ABC, abstractmethod
from dataclasses import dataclass, asdict
from typing import Dict, Any, Tuple

@dataclass
class FittedParams:
    """Container for all parameters needed by the simulator."""
    decision_weights: Dict[str, float]          # e.g., b0,
        b_prev, wF, wW, wC, b_info, b_risk, etc.
    layer_weights: Dict[str, float]             # e.g.,
        family, work_school, community
    info_params: Dict[str, float]               # e.g.,
        campaign_intensity, gamma_info, memory_decay
    noise_params: Dict[str, float]              # e.g.,
        temperature
    meta: Dict[str, Any]                        # e.g.,
        seed, calibrator_name, training_window, notes
    def to_dict(self) -> Dict[str, Any]:
        return asdict(self)

class Calibrator(ABC):
    """Pluggable calibrator interface with a stable evaluation
        callback signature."""
    @abstractmethod
    def fit(self, bundle, simulator, evaluator, train_window:
        Tuple[int, int], seed: int) -> FittedParams:
        """Return FittedParams, fitted strictly on the training
            window."""
```

   - Provide at least two concrete implementations and a string-keyed registry, they must use the evaluator on the training window as its objective, support a budget (iterations), and return the best FittedParams: - LogitHeadCalibrator: fits a logistic decision head from micro-transitions on days_train (L2 regularized; intercept not regularised). - RandomSearchCalibrator: black-box search over selected simulator params (e.g., layer weights, info rates, memory, temperature).

```
       Template:
   1  CALIBRATOR_REGISTRY = {{
   2      "logit_head": LogitHeadCalibrator,
   3      "random_search": RandomSearchCalibrator,
   4  }}
   5
   6  def get_calibrator(name: str, config_path: str | None):
   7      if name not in CALIBRATOR_REGISTRY:
   8          raise ValueError(f"Unknown calibrator: {{name}}")
   9      # Load optional config (JSON/YAML) into kwargs; fall back
               to sensible defaults
  10      ...
  11      return CALIBRATOR_REGISTRY[name](**kwargs)

       4.  Provide an evaluator with the exact callable:
       Template:

   1  def evaluate_params(simulator, params: FittedParams, window) ->
           Dict[str, Any]:
   2      """
   3      Apply `params`, run a forward simulation on `window`, and
           return a metrics dict
   4      containing at least: 'RMSE_aggregate', 'MAE_aggregate', '
           Brier',
   5      'TransitionFit' (with P01, P11, P10, P00).
   6      """

       – Calibrator.fit must only use this callback for scoring on the
       training window.
```

**Prompt:** *This is the prompt for the **Code Generation Agent**.*

## J.3 CODE PATCH PROMPT

When the initial simulator components are constructed using the **Code Generation Prompt** (Appendix J.2), we explicitly require the Code Generation Agent (CGA) to generate a generic calibrator interface. Building on this interface, the CGA is then guided by this **Code Patch Prompt** to extend the implementation by adding the two core SOCIA calibrators: Bayesian Optimization (BO) and Simulation-Based Inference (SBI).

---

**Code Patch Prompt**

You are a Code Generation Agent (CGA). You have already produced a
base code snippet code.

Now, you need to read the existing code and modify it to add a new
calibration method: **SBI (Simulation-Based Inference)**. We will build
the SBI calibrator step by step.

    0. **Add a new SBI calibrator interface**.

        • Create a new calibrator option called SBI.
        • For now, implement the method as a placeholder (pass) but
          ensure that switching between calibration methods (e.g.,
          BO, EA, SBI) is supported.

    1. **Define priors for calibratable parameters**.

        • When using the SBI calibrator, assign a prior distribution
          to each parameter.
        • Each parameter should have a reasonable lower and upper
          bound.

- The prior distribution is a **uniform distribution** within these bounds.
- Implement this without affecting the functionality of other calibrators.

2. **Construct training data for SBI.**
   - For each SBI training run, sample $N = 1000$ parameter sets from the defined priors.
   - For each sampled parameter set, run the simulator once and collect a trajectory.
   - Collect the pairs $(\omega^{(i)}, Y^{(i)})$, where $\omega^{(i)}$ are parameters and $Y^{(i)}$ is the trajectory vector.

3. **Define observables and trajectories.**
   - Example (mask adoption task):
     - Start with $K = 1$: daily average mask adoption rate.
     - A trajectory is a sequence of these observables over all time steps.
     - Flatten the resulting $T \times K$ matrix into a fixed-length vector of dimension $T \cdot K$.
   - Extended observables ($K = 5$):
     - In addition to daily adoption rate, compute daily transition rates between mask states:

       $$P_{01} : \text{non-mask} \rightarrow \text{mask}$$
       $$P_{11} : \text{mask} \rightarrow \text{mask}$$
       $$P_{10} : \text{mask} \rightarrow \text{non-mask}$$
       $$P_{00} : \text{non-mask} \rightarrow \text{non-mask}$$

     - Collect these per day (for $t = 2 \dots T$), forming a $(T - 1) \times 4$ matrix.
     - Combine with adoption rate into a $T \times 5$ matrix (fill day 1 with dummy values if needed).
     - Flatten into a vector of length $T \cdot 5$.

4. **Train a Neural Posterior Estimator (NPE).**
   - With $N = 1000$ pairs of (parameters, trajectories), train a neural posterior estimator.
   - Use the negative conditional log-likelihood loss:

     $$L = -\sum_i \log q_\phi(\omega^{(i)} \mid Y^{(i)}).$$

   - Implement using the existing sbi library.
   - Support density estimators such as **Masked Autoregressive Flow (MAF)** or **Neural Spline Flow (NSF)**.

5. **Posterior saving and checkpointing.**
   - Save the learned posterior distribution and model checkpoints into the folder: mask_adoption_data/outputs_SBICalibrator/.
   - Ensure that checkpoints can be loaded at test time to re-generate the posterior and sample parameters.

This way, the SBI calibrator will be integrated into the simulator alongside existing calibrators, support prior specification, trajectory processing, posterior learning, and checkpointed outputs.

You have already generated the base code {code}. Now, you need to read this code and modify it by adding a new **BO** calibration method.

0. **Add a new calibrator: BoCalibrator.**

- Extend the existing code by implementing a new class or module called BoCalibrator.
- For now, implement its methods with pass as placeholders, but ensure that the calibration workflow can switch between existing methods and BoCalibrator.

1. **Step 1 – Analyze the calibration objective**.
   - Bayesian Optimization (BO) requires an error metric $y$ computed for each parameter set $\omega_i$.
   - Check whether the evaluation metrics already implemented in {code} (e.g., RMSE, MAE, Brier, TransitionFit, etc.) are suitable as the BO optimization objective.
   - If multiple metrics are combined, ensure consistent directionality:
     – Error-type metrics (RMSE, MAE, Brier) are minimized directly.
     – Fitness-type metrics (e.g., TransitionFit, where higher is better) must be inverted, e.g.,

   $$TF\_loss = 1 - \texttt{TransitionFit}.$$

   - Optionally normalize or rescale metrics when combining them into a composite loss.

2. **Step 2 – BO algorithm design.** Implement Bayesian Optimization to tune the calibration parameters $\omega$:
   (a) **Initialization:** Randomly sample several parameter sets $\omega_1, \omega_2, \ldots$ from the search space. Run the simulator on each and collect corresponding errors $y_1, y_2, \ldots$, forming the dataset $\{(\omega_i, y_i)\}$.
   (b) **Gaussian Process (GP) surrogate:** Fit a GP to map parameters $\omega$ to error $y$. The GP should return both mean prediction $\hat{y}$ and uncertainty (confidence intervals).
   (c) **Acquisition function:** Use an acquisition function (e.g., Expected Improvement, Upper Confidence Bound) to balance exploration vs. exploitation. Select the next parameter $\omega_{t+1}$ accordingly.
   (d) **Simulation update:** Run the simulator at $\omega_{t+1}$, obtain the true error $y_{t+1}$, and add $(\omega_{t+1}, y_{t+1})$ back into the dataset.
   (e) **Iterate:** Repeat steps 2–4 until the computational budget is exhausted. Select the parameter set $\omega^*$ with the lowest observed error as the best calibrated result.

3. **Implementation requirements**.
   - Use the BoTorch library to implement Bayesian Optimization with Gaussian Processes.
   - Integrate it into the existing simulator workflow from {code}.
   - Ensure the calibrated parameters can be saved for later testing.
   - Save the best parameters into the output directory {folder1}, using the same format as {file1}, {file2}.

**Prompt:** *This is the code patch prompt for the **Code Generation Agent**.*

## J.4 SIMULATION EXECUTION PROMPT

**Simulation Execution Prompt**

```
You are the Simulation Execution Agent in a system that generates
social simulations.  Your job is to run the generated simulation code
and collect the results.

Task Specification:
{task_spec}

Code Path:
{code_path}

Data Path (if available):
{data_path}

Please execute the simulation code and provide the results.
Consider:
1.  Running the simulation with appropriate parameters
2.  Collecting metrics and outputs
3.  Capturing any runtime errors or issues
4.  Generating visualizations if appropriate

Please structure your response as a JSON object with the following
format:

{
  "execution_status": "success|partial_success|failure",
  "runtime_errors": [
    {
      "error_type": "Type of error",
      "message": "Error message",
      "location": "Where the error occurred"
    }
  ],
  "performance_metrics": {
    "execution_time": time_in_seconds,
    "memory_usage": memory_in_mb
  },
  "simulation_metrics": {
    "metric1": value1,
    "metric2": value2
  },
  "time_series_data": [
    {
      "time_step": step_number,
      "metrics": {
        "metric1": value1,
        "metric2": value2
      }
    }
  ],
  "visualizations": [
    {
      "type": "Type of visualization",
      "path": "Path to the saved visualization",
      "description": "Description of what the visualization shows"
    }
  ],
  "summary": "Summary of the simulation execution and results"
}
```

**Prompt:** *This is the prompt for the **Simulation Execution Agent**.*

## J.5 FEEDBACK GENERATION PROMPT

---

**Feedback Generation Prompt**

You are the Feedback Generation Agent in a system that generates
social simulations. Your job is to synthesize the results of
execution and evaluation to produce *actionable, code-level structural
feedback* for improving the simulator.

Blueprint:
{blueprint}

Current Calibrated Parameters ($\omega$):
{parameter_summary}

Current Code:
{code_content}

Code Changes From Previous Iteration:
{code_diff}

Diagnostic Vector $\delta(\lambda,\omega)$ (per-metric / per-phase residuals and scalar
validation loss):
{diagnostics}

Outer-Loop History (short log of previous iterations, including
structures, diagnostics, losses, and feedback):
{history_log}

SPECIAL REQUIREMENTS:
- At the end of the file, include a direct call to the main()
function (e.g., # Execute main for both direct execution and sandbox
wrapper
invocation followed by main()) instead of using the traditional if
__name__ == "__main__" guard. This is a STANDARD REQUIREMENT for all
simulations in this system and should NOT be considered an issue.

Carefully reflect on how to improve the above simulation **code and
structure** so that the **validation loss decreases**.
Explain your reasoning step by step.
Only propose suggestions that can be directly mapped to **code changes**
(e.g., adding/removing modules, modifying equations, changing
parameterizations); **avoid vague high-level comments**.
Whenever possible, explicitly refer to the diagnostics (including
per-metric and per-phase errors) and the optimized parameter values,
and explain: - which aspect of the dynamics is currently misfit; -
which module / subroutine / variable should be structurally modified.
Your output should be a **numbered list of suggestions**, where each item
corresponds to a **concrete code-editing action**.

Please perform a detailed analysis of the current code, the recent
changes, and the diagnostic vector. Your analysis should:
1. Analyze the current code and structure for:
- Potential logical errors in the implementation (e.g., empty
collections, disconnected components)
- Specific errors in the code (e.g., incorrect type handling, missing
connections between objects)
- Potential execution flow issues that could cause failures or
incorrect results
- Appropriateness of data structures and algorithms for the
simulation task
- Structural mismatches suggested by the diagnostics (e.g.,
systematic bias in certain phases or subgroups)
- Problematic dependencies
2. Analyze the recent changes (from the diff) and the history log to
identify:
- Whether these changes might have introduced new issues or bugs

---

```
- How these changes affect the overall code quality and functionality
- If the changes properly address previous iteration's feedback
- Any unintended side effects of the changes
- Which kinds of past edits in the history have improved or failed to
improve the validation loss

Based on your comprehensive analysis of the code, recent changes,
diagnostics, and history, provide detailed feedback for improvement.
Focus on structural, code-level edits that are directly grounded in
the diagnostics.  Consider:
1.  What are the most critical issues to address?
2.  How can the simulation model (its mechanisms, interactions, and
parameterizations) be improved?
3.  How can the code implementation be enhanced to better reflect the
intended dynamics?
4.  What specific changes would lead to better alignment with
real-world data and reduce the observed residuals?
5.  What are the priorities for the next iteration?

For each identified issue, provide:  - A detailed explanation of the
problem
- The impact on simulation results and diagnostics (which metrics /
phases / groups are affected)
- A specific code-level solution with before/after snippets
- If the issue was introduced or influenced by recent changes,
explicitly note this

Include a top-level code_snippets field in the JSON. This should be
an array of objects, each with "file", "before", and "after" keys,
providing minimal before/after code examples that correspond to
entries in code_improvements.

Please structure your response as a JSON object with the following
format:

{
  "summary": "Brief summary of the overall feedback",
  "critical_issues": [
    {
      "issue": "Description of a critical issue",
      "impact": "Why this issue is important",
      "solution": "Proposed solution",
      "introduced_by_changes": true/false
    }
  ],
  "model_improvements": [
    {
      "aspect": "Aspect of the model to improve",
      "current_approach": "Current implementation",
      "suggested_approach": "Suggested implementation",
      "expected_benefit": "Expected improvement from this change"
    }
  ],
  "code_improvements": [
    {
      "file": "File or component to modify",
      "modification": "Suggested code modification "
                      "(should be a concrete, structural edit)",
      "reason": "Why this modification would help "
                "(refer to specific diagnostics when possible)",
      "related_to_recent_changes": true/false
    }
  ],
  "data_alignment_suggestions": [
    {
```

```
      "metric": "Metric to improve",
      "current_gap":"Current difference between simulation and reality",
      "suggestion": "How to reduce this gap"
    }
  ],
  "prioritized_actions": [
    "First priority action (most impactful structural/code edit)",
    "Second priority action",
    "... (ordered by priority)"
  ],
  "additional_comments": "Any other feedback or observations",
  "code_snippets": [
    {
      "file": "simulation.py",
      "before": "original code snippet",
      "after": "suggested code snippet",
      "addresses_recent_changes": true/false
    }
  ],
  "change_analysis": {
    "summary": "Brief analysis of how the recent changes and "
               "proposed edits affect the codebase",
    "positive_impacts": ["Positive impact 1"],
    "negative_impacts": ["Negative impact 1"],
    "suggestions": ["Suggestion to improve the changes"]
  }
}
```

**Prompt:** *This is the prompt for the **Feedback Generation Agent**.*

## K  LIMITATIONS AND FUTURE WORK

While `SOCIA` demonstrates promising capabilities in automating the construction of high-fidelity social simulators, several limitations remain:

1. Although the framework supports diverse simulation tasks and integrates multi-agent co-ordination with feedback-driven refinement, its performance is bounded by the capabilities and biases of the underlying LLM backbone (e.g., GPT-5). In highly novel or underrepresented domains, generated simulators may struggle to generalize beyond memorized patterns or may fail to produce semantically valid behaviors.

2. The current design of `SOCIA` assumes a synchronous and centralized orchestration model via the Workflow Manager, which may pose scalability challenges in extremely large-scale or decentralized environments. Future work could explore distributed coordination strategies or hierarchical agent delegation to enable broader scalability and responsiveness.

3. Our current instantiation focuses on agent-based social simulations. In principle, however, the same agents—especially the SEA/FGA pair—could be instantiated around non-agent-based scientific simulators (e.g., ODE/PDE-based models, physics engines, or climate and epidemiological models) by treating the simulator as a black-box forward model that SEA calibrates and FGA structurally refines (e.g., via configuration files or model components). We view this as a promising direction for extending `SOCIA` beyond social domains, but we do *not* claim immediate, off-the-shelf support for such scientific simulators in the current work.

4. While `SOCIA` can construct aggregate-level simulators with explicit structure and calibration, it does not yet support the creation of *communicative simulation agents*—i.e., simulated entities endowed with LLM-based memory and dialogue context that can engage in conversations and form communities where communication itself is governed by an LLM-driven protocol. This represents a current functional limitation of the framework.

5. We plan to extend `SOCIA`'s capabilities to support more complex scenarios involving long-term planning, multi-agent cooperation and competition, and real-time decision-making under uncertainty. This will require augmenting agents with stronger memory, strategic reasoning, and cross-episode learning capacities. In addition, integrating causal modeling and counterfactual inference remains a key direction to enhance the explanatory power and scientific utility of `SOCIA`-generated simulators.

## L  ETHICS STATEMENT

`SOCIA` involves a human-in-the-loop component, where user feedback can be collected to refine simulator code in real time during iterations. Feedback may also be gathered offline to inform revisions of the overall framework, methodology, or calibration design. Since collecting feedback from human participants raises ethical considerations, this work has undergone review and approval by the Research Ethics & Compliance Support (RECS) office at UNSW Sydney (application ID: iRECS8992). Accordingly, the collection of human feedback within `SOCIA` complies with established ethical requirements.

