# OpenReview forum: "SOCIA: Joint Structure–Parameter Co-Optimization for Automated Simulator Construction"
_ICLR.cc/2026/Conference — Submitted to ICLR 2026_

### Official Review · Reviewer_3qgX · 2025-10-24

**Soundness:** 2
**Presentation:** 2
**Contribution:** 3
**Rating:** 4
**Confidence:** 3

**Summary:**

The article introduces a framework for building and tuning simulation models (primarily agent based models). The chain starts with a chain-of-structure specification to encapsulate the situation being modelled. An LLM is used to encode this framework into code, and a combination of Bayesian optimisation for parameter tuning, and further LLM methods for model updating, are used to fine tune the model. Experiments are given in which, of course, the current method provides higher performance than the baselines.

**Strengths:**

I really liked this article's full spectrum approach to modelling, going all the way from the problem specification to fine tuning and testing the resulting simulation model.
The paper is on a different topic to most machine learning conference papers (at least the ones I read) but uses lots of highly relevant techniques and addresses problems that machine learning researchers address, using a slightly different toolset. I think it would be a strong contribution to the conference.

**Weaknesses:**

There are some gaps in the presentation. Primarily, the key "Algorithm 1" is actually in the Appendix, which I think is verging on cheating the page limit unfairly.
However there were also lots of details of the method that I could not ascertain from reading the article. The most frustrating one for me was the iterative refinement of model structure. Only one sentence used to present this (lines 314-315) but it is a complex and interesting part of the method.

**Questions:**

Please can you give some more detail on each of the components of your method (but not the Bayesian optimisation, which is completely standard). I think they're all interesting but will struggle to accept a paper when so much of the method is not described at all.

---

> ### Author Response · Authors · 2025-11-29
> **Response to Reviewer 3qgX**
>
> > **Q1**: Primarily, the key "Algorithm 1" is actually in the Appendix, which I think is verging on cheating the page limit unfairly.
>
>  **A1**: Our intention was not to game the page limit, but to keep the main text lighter and move the full pseudo-code to the appendix. Section 3.5 already describes the structure–parameter refinement loop in prose, and all the explanation of the key steps of Algorithm 1 (objective, update rule, and stopping criterion) can be viewed in the revised ```Appendix A```.
>
> ---
>
> All other concerns are addressed in the **Consolidated Response to All Reviewers**.
>
> ---

---

### Official Review · Reviewer_AoLv · 2025-10-31

**Soundness:** 2
**Presentation:** 1
**Contribution:** 2
**Rating:** 2
**Confidence:** 2

**Summary:**

The paper proposes an agentic workflow for automatically building multi-agent simulations based on observational data. The main novelty seems to be the joint optimization of the simulator structure and its hyperparameters.

**Strengths:**

n.a.

**Weaknesses:**

[motivation and introduction] While the paper tackles a significant and complex challenge with state-of-the-art methods, the introduction is highly generic. It does not adequately establish what specific task SOCIA performs or what kind of simulators it produces. After multiple readings of the intro, it remains unclear to me what the system’s concrete task and output is (see Figure 1: is SOCIA generating executable code, calibrated data, forecasts, or decision policies?). This becomes a bit clearer after browsing the appendix.
--> Providing an early, concrete example of a representative simulator and a clear definition of the modeling task would give readers a much clearer entry point into the paper.

[readability] The paper’s notation is inconsistent and often undefined, which makes it hard to understand the central methodology. For instance, $\lambda$ is described as a “mechanistic blueprint” (l161) while $B$ is later introduced as a “simulator blueprint” taking $\lambda, \omega$ as input (l212). Furthermore, $T$ denotes a textual task description and a data count (line 183). These ambiguities, combined with very long and nested sentences (e.g., single sentences spanning lines 219-229, 241-247), make the content difficult to follow.
--> Clearer notation and more concise writing would significantly improve readability and make the paper accessible to a broader audience.

**Questions:**

This paper lies somewhat outside my core area of expertise (I was likely assigned since I am very familiar with BO), and I found it difficult to grasp the task definition and central methodology fully (see comments below). As a result, I could not thoroughly review the entire paper and appendix, and my feedback focuses on the introduction, technical problem setup, and methodology. While I recognize the potential relevance of the work, I believe the paper requires a major revision. Even for readers outside the immediate subfield (like me), the introduction should more clearly motivate the task/application and clearly define the problem and main ideas to make the paper accessible to a broader audience.

---

> ### Author Response · Authors · 2025-11-29
> **Response to Reviewer AoLv**
>
> > **Q1**: What specific task SOCIA performs or what kind of simulators it produces? What the system’s concrete task and output is? The introduction should more clearly motivate the task/application and clearly define the problem and main ideas.
>
> **A1**: SOCIA’s concrete task is to take a natural-language modeling brief plus observational data and automatically produce an executable, calibrated multi-agent simulator: the system (i) generates and refines simulator code, (ii) calibrates its parameters against data, and (iii) returns the resulting simulator along with diagnostics and forecast trajectories.
>
>  In the revised manuscript, we make this task and output explicit in the opening paragraphs to give readers a clear entry point before they reach the technical sections (```lines 103-108```).
>
> ---
>
> > **Q2**: The paper’s notation is inconsistent, making it hard to understand the central methodology. λ is described as a “mechanistic blueprint” (l161) while B is later introduced as a “simulator blueprint” taking λ, ω as input (l212). Furthermore, T denotes a textual task description and a data count (line 183).
>
> **A2**: In the revised manuscript, we made the following concrete changes:
>
> *1). Disambiguating $B$ and $\lambda$.* We now use a consistent terminology (in ```lines 231, 246-247, 251, and 255-256```):
>  - $B$ denotes the *task-level blueprint* produced by the DAA (a structured summary of the task, data schema, and modeling choices).
>  - $\lambda$ denotes the *mechanistic structure* of the simulator (agents, topology, transition rules, observables) that is compiled from $B$.
>  - We removed the phrase “simulator blueprint” for $B$ and systematically refer to $B$ as “task blueprint” and $\lambda$ as the “simulator structure” wherever they appear.
>
>  *2). Removing overloaded use of $T$.* We acknowledge that using $T$ both for the textual task description and for a data count was ambiguous. In the revision (```line 199```),
>   - $T$ is used *only* for the natural-language task description;
>   - dataset sizes and time horizons are now denoted by standard numeric symbols (e.g., $N$, $T_{\text{steps}}$​) instead of reusing $T$.
>
>   *3). Clarifying the core variables and notation.* In Section 3.1, we introduce $x_t$ ​ as the system state and $y_t$ ​ as the observables. In Section 3.3, however, we inadvertently followed standard SBI notation and reused $x$ to denote the “observed data,” which creates a notational conflict.
>
> In the revised manuscript, we resolve this ambiguity by redefining the SBI posterior to be conditioned on the observables rather than on $x$: we change the conditioning variable from $x$ to $y_{obs}$ in the SBI description (```lines 207-208, 275–277```), and we explicitly clarify this convention in ```Section 3.1 (lines 187–188)```.
>
> ---
>
> All other concerns are addressed in the **Consolidated Response to All Reviewers**.
>
> ---

---

### Official Review · Reviewer_t7Ju · 2025-11-02

**Soundness:** 2
**Presentation:** 2
**Contribution:** 3
**Rating:** 2
**Confidence:** 4

**Summary:**

The paper presents SOCIA, a framework for joint structure–parameter co-optimization of
simulators using LLM-based agents. The system orchestrates a pipeline consisting of a
Chain-of-Structure (CoS) generator, a Code Generation Agent (CGA), a Simulation
Execution Agent (SEA) for calibration via Bayesian Optimization (BO) and
Simulation-Based Inference (SBI), and a Feedback Generation Agent (FGA) for structure
refinement. The overall goal is to automate the design, implementation, and calibration of
simulation models, particularly agent-based models, by coupling structural synthesis
with parameter learning and uncertainty estimation. Experiments span three tasks, user
modeling, mask adoption, and personal mobility, evaluating both in-distribution (ID) and
out-of-distribution (OOD) regimes. SOCIA variants outperform or match existing LLM-based
baselines (e.g., GSIM, AI Scientist). The work positions itself as a holistic framework for simulation-based science and a
step towards self-calibrating simulation agents.

**Strengths:**

- **Ambitious and holistic framing.** SOCIA tackles an important and underexplored
  question: how to couple structural design and parameter inference in simulation-based
  modeling using AI agents. The modular pipeline with separate agents working on different parts of the pipeline is
  well-motivated.
- **Novel integration.** The combination of LLM-driven structure synthesis
  with parameter tuning via both GP-based BO and SBI is a novel and interesting
  direction. The use of SBI for posterior estimation and OOD robustness is particularly
  compelling.
- **Empirical coverage.** The experiments span multiple tasks (ID, OOD, intervention),
  and ablation results illustrate that removing CoS or calibration steps substantially
  degrades performance.
- **Clear motivation for uncertainty-aware calibration.** The contrast between SOCIA-BO
  and SOCIA-SBI provides useful insight into where each approach excels, and supports
  the authors’ argument that uncertainty helps under regime shifts.
- **Potentially impactful vision.** If properly extended and validated, the system could
  form the basis for practical LLM-assisted modeling workflows—especially for
  agent-based social or behavioral simulations.

**Weaknesses:**

### Conceptual and methodological clarity

- **Implicit focus on agent-based models.** Although the abstract and introduction
  suggest generality (“simulation-based modeling”), the formulation, notation, and
  examples are clearly tailored to ABMs (agents, policies, exogenous inputs). The paper
  should explicitly state this scope and discuss how (or whether) SOCIA can generalize
  to domains with complex scientific simulators (e.g., neuroscience, physics).
- **Technical definitions remain vague.** Core components in Section 3.1–3.2 are
  underdefined: the aggregation operator $A$, the objectives $J_{\text{train}},
  J_{\text{val}}$, and metrics like CRPS appear without formal definitions. The
  relationship between variables $x, y, u, \omega$ is unclear across sections (e.g.,
  the SBI posteriors are defined conditioned on x, not y).
- **Ambiguity between “agents.”** The term *agent* refers both to simulated entities and
  to LLM components orchestrating the workflow. This dual meaning often causes confusion
  and should be disambiguated consistently (e.g., “simulation agent” vs “AI agent”).

### Empirical evaluation and baselines

- **Missing comparison to standard manual workflows.** Evaluation focuses on LLM-based
  systems (GSIM, AI Scientist) but omits the central baseline of *human-designed
  simulators calibrated with BO or SBI*. This is critical to assessing whether LLM-based
  orchestration meaningfully assists existing modeling pipelines.
- **No expert validation of generated simulators.** Results rely exclusively on
  numerical metrics. It would be informative to include domain-expert evaluation of
  simulator plausibility (e.g., whether generated structures make physical or behavioral
  sense).
- **Limited compute transparency.** The paper does not report simulator call counts,
  wall-clock time, or resource usage for BO and SBI calibration loops. Given that
  structural edits can trigger full re-runs of calibration, compute requirements are
  important for assessing scalability and practical feasibility.
- **Section 4.3 lacks orientation and purpose clarity.** The section introduces
  additional experiments that modify the main tasks, but it lacks a short introductory
  statement explaining *why* these experiments are run (e.g., testing generalization,
  intervention robustness, or pipeline autonomy).
- **Related work coverage is narrow.** The paper cites task-specific SBI/BO works but
  omits standard references such as Cranmer et al. (2019) or general introductory papers
  like Deistler & Boelts (2025), which would help situate the work for non-specialist
  readers.

### Presentation and structure

- The introduction and Section 3 remain abstract; a running example or concrete case
  study would help anchor the reader.
- Figure 1 is referenced late (p. 6) despite being essential for understanding the
  architecture—earlier guidance to it would improve readability.
- Minor confusion arises from describing SBI as “simulate–compare–learn”, which reads
  like ABC but actually describes conditional density estimation via NPE.

### Overall contribution

The main contribution of the paper lies in system integration rather than algorithmic novelty. SOCIA presents an ambitious and well-engineered orchestration of LLM-based structure generation, simulator code synthesis, and calibration via standard BO and SBI components. This is a valuable step toward automating simulation-based modeling workflows, particularly for agent-based settings.

However, the paper does not yet articulate a clear technical algorithm underlying the structure–parameter co-optimization loop. The proposed “structural refinement” process appears to operate heuristically, e.g., driven by diagnostics and LLM proposals, without a formally specified objective, acceptance rule, or convergence behavior. While this is understandable given the non-differentiable nature of the involved components, the lack of algorithmic detail makes it difficult to assess the method’s reliability or theoretical grounding.

In its current form, SOCIA should thus be viewed primarily as a proof-of-concept system demonstrating how modern language models can coordinate established inference and optimization techniques. The idea is promising and potentially impactful, but the technical contribution would be significantly strengthened by a more precise definition of the optimization loop, explicit evaluation traces of structure edits, and a discussion of what (if anything) can be guaranteed or bounded within this framework.

**Questions:**

1. **Aggregation operator \(A\):** How exactly is \(A\) defined and implemented (line
   188)? Does it perform statistical aggregation or empirical mapping from micro-level
   simulations to macro-level indicators?
2. **Objectives and metrics:** What is $J$? How is CRPS computed and used in the loss?
   (line 191)
3. **Notation consistency:** Posteriors are defined as $p(\omega\mid x)$, but $x$ is
   described as system state while training data involve $(y,u)$. Please clarify the
   variable roles.
4. **Initialization heuristics:** (line 320) What heuristics justify assuming that CoS
   provides a near-optimal starting point? How robust is this when the constructed
   simulator poorly fits the data?
5. **Calibration strategy:** How does the CGA or SEA decide between running BO and SBI?
   Are both always executed, and if so, how are point and posterior estimates combined?
6. **Compute scaling:** Each structural edit triggers new simulations and retraining of
   BO/SBI models. What are the wall-clock costs, simulator call budgets, and
   computational resources per task?
7. **Posterior sampling:** SBI inference reportedly draws 50–100 posterior samples. Why
   such a low number, given that NPE allows essentially free sampling?
8. **Baselines:** How exactly do the Random Search and LR baselines operate? Which
   simulator are they tuning?

---

> ### Author Response · Authors · 2025-11-29
> **Resonse to Reviewer t7Ju -- Part 1**
>
> > **Q1**: What heuristics justify assuming that CoS provides a near-optimal starting point? How robust is this when the constructed simulator poorly fits the data?
>
>  **A1**: Our design is: the outer refinement loop is explicitly budgeted to at most 5 iterations, with a patience hyperparameter $K=2$ for strict-improvement steps in the validation loss. Empirically, across our experiments the best validation loss saturates within **4–5 outer iterations**, with the patience criterion already triggered by that point. This pattern is consistent with CoS providing a **high-quality starting blueprint**: the initial structure derived from CoS is already close to a good solution in the diagnostic space, so only a small number of targeted refinements are needed to reach a plateau of “near-optimal” structures under our objective.
>
> When multiple structures yield comparably good diagnostics, we do not attempt to break this equivalence via unjustified assumptions; instead, we apply a simple Occam-style heuristic and prefer the more compact simulator (fewer mechanisms / shorter code) among those with similar validation loss. Thus, CoS is not taken as an oracle but as a strong, domain-informed prior that empirically places the search in a favorable region of the structure space, as evidenced by the rapid saturation of improvements within our 5-step budget, while the refinement loop and selection heuristic handle residual mismatch and non-uniqueness rather than presuming exact recovery of a single “true” simulator.
>
> In the revised manuscript, we address this point in the ```Ablation findings paragraph```, ```lines 513–519```.
>
> ---
>
> > **Q2**: Is SOCIA currently focused only on ABMs? Whether and how it could extend to other complex scientific simulators.
>
>  **A2**: Our _current_ instantiation of SOCIA is indeed primarily targeted at agent-based and micro-simulation settings, and we will make this scope explicit in the introduction to avoid over-claiming generality. At the same time, the core design of SOCIA separates a generic orchestration pattern (task-to-blueprint translation, code generation, diagnostic-driven structure–parameter refinement) from its ABM-specific realization: in principle, the same loop can operate over any simulator that (i) exposes a parameterized structure representation and (ii) can be evaluated via trajectory-level diagnostics.
>
> In the revised manuscript, we (i) clearly state that the experiments and notation focus on ABMs (```lines 129–132```), and (ii) add a short subsection discussing how the same agents (SEA/FGA) could be instantiated for non-agent-based scientific simulators, while also acknowledging that such extensions are future work rather than claims of immediate, off-the-shelf support (```Appendix K```).
>
> ---
>
> > **Q3**:  In Section 3.1–3.2, the aggregation operator A, the objectives J_train, J_val, and metrics like CRPS are underdefined.
>
> **A3**: In our notation , $A(\cdot)$ is a _statistical aggregation operator_ that maps micro-level simulation outputs (e.g., each agent’s trajectories, ratings, or binary mask-wearing decisions) to macro-level indicators (e.g., daily adoption rates, aggregate star-rating distributions), which are then compared to their ground-truth counterparts.
>
> $L$ is instantiated in all of our experiments using the same task-specific discrepancy metrics we report in the results (e.g., aggregate RMSE/MAE, Brier score, TransitionFit on the aggregated trajectories), and $\Psi$ encodes hard/soft domain constraints.
>
> CRPS in this paragraph was intended only as an example of a standard probabilistic scoring rule that _could_ be used when full predictive distributions are available; in the experiments reported in the paper we in fact *do not use CRPS at all*, because we work with point predictions and their derived summary metrics.
>
> In the revised manuscript, ```lines 203-214```, we (i) explicitly define $A(\cdot)$ and $J$​ and (ii) remove the mention of CRPS, so that the objectives and metrics are precisely specified and aligned with what is actually implemented.
>
> ---
>
> > **Q4**: The relationship between variables x, y, u, w, is unclear across sections.
>
> **A4**: In Section 3.1, we introduce $x_t$ ​ as the system state and $y_t$ ​ as the observables. In Section 3.3, however, we inadvertently followed standard SBI notation and reused $x$ to denote the “observed data,” which creates a notational conflict. In the revised manuscript, we resolve this ambiguity by redefining the SBI posterior to be conditioned on the observables rather than on $x$: we change the conditioning variable from $x$ to $y_{obs}$ in the SBI description (```lines 207-208, 274–276```), and we explicitly clarify this convention in ```Section 3.1 (lines 187–188)```.
>
> ---

---

> ### Author Response · Authors · 2025-11-29
> **Resonse to Reviewer t7Ju -- Part 2**
>
> > **Q5**: Ambiguity between “agents”: the term agent refers both to simulated entities and to LLM components orchestrating the workflow.
>
> **A5**: In the revised manuscript, we perform a global disambiguation of the term “agent.” For entities inside the simulator, we now consistently use **“simulation agents”** (see ```lines 106, 177, 220, 221```). For the workflow components DAA/CGA/SEA/FGA, we refer to them as **“LLM orchestration agents”**, and we explicitly clarify this terminology at their first introduction (```lines 106-108, 255```).
>
> ---
>
> > **Q6**: Related work coverage is narrow.
>
> **A6**: In the revised manuscript, we add the missing reference Deistler & Boelts (2025) (```line 275```), but *Cranmer et al. (2019)* is already cited.
>
> ---
>
> > **Q7**: Minor confusion arises from describing SBI as “simulate–compare–learn”, which reads like ABC but actually describes conditional density estimation via NPE.
>
> **A7**: In the original manuscript, the Introduction and Section 4.3 (User modeling) described SBI as “simulate–compare–accept” to mirror our calibration workflow, where we simulate predictions, compare them to ground truth, and accept a configuration if fidelity is adequate (described in  Section 3.5). We agree this wording resembles the classic ABC procedure and may cause confusion.
>
> In the revised manuscript, we instead use a standard NPE-style description—“to learn a posterior distribution” (```lines 119 and 473```)—to more accurately reflect our NPE-based implementation.
>
> ---
>
> > **Q8**: How to decide between running BO and SBI? Are both always executed, and if so, how are point and posterior estimates combined?
>
>  **A8**: In SOCIA, BO and SBI are **alternative** calibration schemas, not two stages that are always run and then merged.
>
> When a task only requires a *point estimate* of the parameters, we use BO to directly optimize the scalar validation objective and take the BO optimum as $\omega^*$. When the user or application requires *full posterior information / uncertainty quantification*, we instead run SBI to approximate $p(\omega \mid D, \lambda)$ and then instantiate the simulator with a point drawn from this posterior as $\omega^*$, while retaining the posterior for for uncertainty quantification (UQ).
>
> We do **not** combine BO and SBI outputs in a single run. We expose both options so that users can flexibly choose the calibration method appropriate to their domain rather than always executing and fusing both.
>
> We clarify this design in the revised text (```Section 3.3, lines 286-290```).
>
> ---
>
> > **Q9**: Evaluation omits the central baseline of human-designed simulators calibrated with BO or SBI.
>
>  **A9**: Our experimental focus was on *automated LLM-based systems* (G-SIM, AI Scientist) because SOCIA is primarily proposed as an orchestration layer for such pipelines, and those systems represent the closest _methodological_ baselines: they also start from natural-language task descriptions and automatically generate and calibrate simulators. In contrast, human-designed simulators calibrated with BO or SBI assume that an appropriate model structure has already been hand-specified; this is a **different regime**, and our goal is not to replace domain experts in designing gold-standard models, but to reduce the amount of manual simulator authoring needed to reach a useful, calibrated baseline.
>
> ---
>
> > **Q10**: The paper does not report simulator call counts, wall-clock time, or resource usage for BO and SBI calibration loops. What are the wall-clock costs, simulator call budgets, and computational resources per task?
>
>  **A10**: For each task, we typically need 3–4 outer-loop iterations to obtain stable, high-quality simulator code. The first draft takes ~17 minutes, each refinement ~10 minutes, so a full optimization run is ≈40 minutes wall-clock, with a fixed small set of LLM calls (planner/critic/editor/verifier) per iteration and a total budget of about 1M tokens per task. In the revised paper, we add these numbers to the paper for transparency (```lines 419-423```).
>
> ---

---

> ### Author Response · Authors · 2025-11-29
> **Resonse to Reviewer t7Ju -- Part 3**
>
> > **Q11**: Section 4.3 lacks a short introductory statement explaining why these experiments are run
>
>  **A11**: Our intention was _not_ to introduce unrelated extra experiments, but to provide **qualitative case studies** that showcase specific capabilities of SOCIA-generated simulators using controlled modifications of the main tasks.
>
> In particular, section 4.3 demonstrates that SOCIA can: (i) incorporate additional parameters to correct black-box, one-step LLM scores that have no roll-out dynamics; (ii) represent exogenous signal interventions that directly modulate agents’ decisions; and (iii) handle differences between in-distribution and out-of-distribution prediction regimes within the same simulator. These analyses are meant to illustrate the flexibility of the generated simulators across different simulation environments.
>
> In the revision, we add a short introductory paragraph at the start of Section 4.3 explicitly stating this goal (```lines 467-468```).
>
> ---
>
> > **Q12**: SBI inference reportedly draws 50–100 posterior samples. Why such a low number, given that NPE allows essentially free sampling?
>
>  **A12**: We appreciate the reviewer’s concern about the reported number of posterior samples. Our original wording was indeed misleading, and we will clarify the actual settings and the rationale.
>
> First, for tasks **without any LLM calls in the simulator** (Mask Adoption and Personal Mobility), we already draw **2,000** posterior samples from the NPE posterior for each configuration; these tasks are purely simulator-based and sampling is computationally cheap.
>
> The lower sample counts only apply to the *User Modeling* task, where the simulator contains an embedded LLM call: after the LLM predicts a star rating, the simulator applies a learned correction based on calibrated parameters. In this setting, naively evaluating many posterior samples would incur very high token and latency costs, because each sample would trigger a full LLM call for every test point.
>
> To mitigate this, we exploit a structural property of the SOCIA-generated simulator: for a given test point, the *LLM prompt is fixed*, so the raw LLM prediction is also fixed across all posterior samples. We therefore introduce an **LLM cache**: the LLM is called once per test point, and all posterior samples only change the correction parameters, not the underlying LLM output. With this cache, we can in principle draw arbitrarily many posterior samples without additional LLM cost.
>
> After introducing the cache, we ran a sensitivity study on the User Modeling task to quantify how the number of posterior samples affects predictive performance (MAE; mean ± 95% CI):
> | # of samples | MAE ↓         |
> |-------------:|---------------|
> | 50           | 0.6228 ± 0.03 |
> | 100          | 0.6426 ± 0.08 |
> | 200          | 0.6249 ± 0.04 |
> | 500          | 0.6280 ± 0.06 |
> | 1000         | 0.6312 ± 0.05 |
> | 2000         | 0.6322 ± 0.04 |
> | 5000         | 0.6301 ± 0.06 |
>
> We observe that MAE stabilizes for *500–5000 samples*, with variations well within the confidence intervals, suggesting that the posterior has effectively converged and additional samples mainly reduce Monte Carlo noise. The results with 50–100 samples are already within the same error range and **do not materially change the conclusions**. In the revision, we (i) clarify that all tasks use 2,000 posterior samples (```lines 388, 414-415```), (ii) describe the LLM-cache mechanism for the User Modeling task (```lines 414-418```), and (iii) report the SOCIA-SBI's result for the *User Modeling* task using 2,000 posterior samples (in ```Table 1```).
>
> ---
>
>
> > **Q13**: How exactly do the Random Search and LR baselines operate? Which simulator are they tuning?
>
>  **A13**: As already written in the main text, both *Random Search* and *LR* are **pure parameter-calibration baselines applied to the same fixed simulator structure** as SOCIA, using exactly the same validation objective $J$ and diagnostics.
>
> -   *Random Search* samples parameter vectors from the same prior ranges used by SOCIA, evaluates the simulator under $J_{\text{val}}$​ for each sample, and reports the best-performing parameter setting found within the budget as the baseline.
>
> -   *LR* is a simple surrogate-based calibration scheme: it uses a lightweight linear model over previously evaluated parameter–loss pairs to guide the next proposals in parameter space, again always tuning the same underlying simulator structure.
>
> We revise ```lines 397-402``` to make the description of using *Random Search* and *LR* more clearly.
>
> ---
>
> All other concerns are addressed in the **Consolidated Response to All Reviewers**.
>
> ---

---

### Author Response · Authors · 2025-11-29
**To all reviewers -- A revised version of our submission incorporating all rebuttal updates has been uploaded.**

Dear Reviewers,

Thank you sincerely for the time, care, and constructive dialogue you invested during the rebuttal period. Your detailed feedback has been invaluable in clarifying our contributions and highlighting concrete avenues for improvement. These insights strengthen the present submission and help us better align with the standards and expectations of the ICLR community.

In response to your comments, we have addressed each point in the rebuttal and incorporated the corresponding revisions into the paper; the **revised manuscript has been uploaded**. For each issue, our rebuttal (i) restates and clarifies the concern, (ii) explains our resolution, and (iii) **cites the exact locations** in the revised manuscript (**highlighted in blue**) where the changes were made, so you can quickly verify the edits.

For the same questions raised by multiple reviewers, we provide a **consolidated response** to avoid unnecessary repetition and keep the rebuttal concise. The detailed rebuttal follows.

---

> ### Author Response · Authors · 2025-11-29
> **Consolidated  Response to all reviewers -- Part 1**
>
> **Response to all reviewers**
> > **Q1**: The loop lacks a clearly defined objective, update rule, and convergence behavior.
>
> **A1.** The outer structure–parameter refinement loop in SOCIA is defined as a nested optimization procedure with an explicit objective, a greedy update rule, and a well-specified convergence/termination mechanism. The **objective** is to minimize a validation loss that aggregates task-specific diagnostics: for each structure–parameter pair $(\lambda,\omega)$ and dataset $D$, we define $J(\lambda,\omega;D)$, and the outer loop seeks to decrease $J$ over successive refinements of the simulator. Given a current best configuration $(\lambda^{\ast},\omega^{\ast})$ with best-so-far loss $J^{\ast} = J(\lambda^{\ast},\omega^{\ast};D)$, each outer iteration applies a _greedy update rule_: the Feedback Generation Agent proposes a set of code-level structural edits to the current simulator's structure, the Simulator Evaluation Agent re-calibrates $\omega$ for this modified simulator, and the resulting pair $(\lambda_{\text{new}},\omega_{\text{new}})$ is _accepted_ only if it yields a strictly lower validation loss, $J_{\text{new}} = J(\lambda_{\text{new}},\omega_{\text{new}};D) < J^{\ast}$.
>
> We set the tolerance to zero in practice; because each $J(\lambda,\omega)$ is computed from fixed random seeds and averaged over multiple rollouts, this strict-improvement rule is numerically stable, while a patience hyperparameter absorbs residual noise. Rather than fixing a global numerical threshold $\varepsilon$ (which would be hard to choose uniformly across tasks with different scales and metrics), we use an _implicit $\varepsilon$ via patience and a hard budget on outer iterations_: the loop maintains a counter of consecutive outer iterations with no accepted improvement of $J^{\ast}$ and terminates once no accepted edit has improved $J^{\ast}$ for $K=2$ consecutive outer iterations, or when a preset budget of at most _5_ outer iterations is exhausted. Under this design, (i) the sequence of best-so-far losses $J^{\ast}$ is monotonically non-increasing by construction; (ii) the algorithm is guaranteed to terminate in a finite number of outer iterations (bounded by the budget); and (iii) termination adaptively occurs once further simulator refinements cease to yield meaningful diagnostic improvements, without relying on an arbitrary global $\varepsilon$.
>
> In the revision, we have integrated the rebuttal content into the main text (```Section 3.5 Iterative Refinement```), ```Appendix A```, and ```Algorithm 1 (the pseudocode)```.

---

> ### Author Response · Authors · 2025-11-29
> **Consolidated  Response to all reviewers -- Part 2**
>
> > **Q2**: The structure–parameter co-optimization loop is underexplained; The structural refinement process appears heuristic and ad hoc.
>
> **A2**: We agree that the original draft did not spell out the loop in enough detail. However, the procedure is in fact systematic and driven by explicit diagnostics, not ad hoc edits.
>
> First, the *Simulator Evaluation Agent (SEA)* defines the _inner_ parameter-calibration step for a fixed blueprint $\lambda$. Given calibrated parameters $\omega$, SEA runs multiple roll-outs under fixed random seeds and computes a *vector* of task-specific discrepancies $\delta(\lambda,\omega)$: this includes the overall validation loss $J(\lambda,\omega;D)$ and its decomposition into per-metric errors (e.g., trajectory-based statistics such as SD/SI/DARD/STVD), as well as subgroup- and time-resolved residuals (e.g., errors for different user cohorts or temporal phases). This diagnostic vector plays two roles: (i) the scalar component $J$ is used for *greedy acceptance and early stopping* in the outer loop (with an implicit threshold and patience $K=2$ as described elsewhere), and (ii) the finer-grained components of $\delta$ provide *structured evidence about _where_ and _how_ the current simulator fails*, which then drives structural refinement.
>
> Second, structural refinement is implemented by the *Feedback Generation Agent (FGA)* as a constrained _policy_ over blueprints, not an unconstrained heuristic. In each outer iteration, the FGA prompt includes: (i) the task blueprint obtained from the chain-of-structure representation (system/task description), (ii) the current simulator code for this iteration, (iii) the optimized parameter vector $\omega$, (iv) the full diagnostic vector $\delta(\lambda,\omega)$ from SEA (including metric-wise and time-/group-resolved errors), and (v) a short history of previous code-edit suggestions and their resulting validation losses as in-context examples. On top of this context, the FGA is instructed to produce only *actionable, code-level structural edits*, explicitly grounded in the diagnostics, for example:
> ```
> Carefully reflect on how to improve the above simulation code so that the validation loss decreases.
>
> Explain your reasoning step by step.
>
> Only propose suggestions that can be directly mapped to code changes
> (e.g., adding/removing modules, modifying equations, changing parameterizations);
> avoid vague high-level comments.
>
> Whenever possible, explicitly refer to the diagnostics (including per-metric
> and per-phase errors) and the optimized parameter values, and explain:
> - which aspect of the dynamics is currently misfit;
> - which submodule / variable should be structurally modified.
>
> Your output should be a numbered list of suggestions, where each item
> corresponds to a concrete code-editing action.
> ```
>
> The resulting list of structural edit suggestions is logged in the *historic log* and fed back as context in subsequent iterations, so the FGA does not “start from scratch” but learns, in-context, which kinds of edits have previously improved or failed to improve $J$. Combined with the *SEA*-driven diagnostics and the greedy acceptance + patience-based stopping rule in the outer loop, this yields a *well-defined structure–parameter co-optimization procedure*: (i) SEA calibrates $\omega$ and produces a rich diagnostic vector $\delta(\lambda,\omega)$; (ii) FGA maps $(\lambda,\omega,\delta,\text{history})$ to a small, interpretable set of structural edits; (iii) only edits that strictly improve the validation loss are accepted, and the loop terminates once no accepted edit improves the best-so-far $J$ for two consecutive outer iterations.
>
> In the revision, we have integrated the rebuttal content into the main text (```Section 3.5 Iterative Refinement```), ```Appendix A```, and ```Algorithm 1 (the pseudocode)```. Also, the FGA prompt can be reviewed in ```Appendix J.5```.

---

### Author Response · Authors · 2025-11-30
**Respectful Summary of Reviewer Feedback for Your Kind Consideration**

Dear Area Chairs and Reviewers,

We thank all reviewers for the thoughtful evaluations and constructive feedback.
Reviewer **t7Ju** highlighted SOCIA’s “ambitious and holistic framing,” its “novel integration” of LLM-driven structure synthesis with BO/SBI calibration, and described it as “a valuable step toward automating simulation-based modeling workflows, particularly for agent-based settings.”
Reviewer **3qgX** wrote that they “really liked this article's full spectrum approach to modelling,” noting that although the topic differs from most ML papers, it “would be a strong contribution to the conference.”
Reviewer **AoLv** emphasized the joint optimization of simulator structure and hyperparameters as the main novelty of our agentic workflow.

----------

### Addressing the main concerns

Crucially, the methodological concerns about the structure–parameter co-optimization loop and the role of BO/SBI have been fully addressed in the revised Sections 3.1–3.5, and the remaining comments focus on experiments and presentation/notation, which we have also clarified and updated.

In the revised paper, we addressed them as follows:

#### Methodology
-   **Structure–parameter refinement loop (objective, update rule, convergence).**
    We now explicitly define the outer loop’s objective, greedy acceptance rule, and stopping criterion.
    → **Revised paper:** `Section 3.5 Iterative Refinement`, `Algorithm 1 (in Appendix A)`, and `Feedback Generation Agent prompt (in Appendix J.5)`.

-   **Role of CoS and robustness of the starting structure.**
    We explain that CoS provides a domain-informed prior and support this with ablation study (SOCIA-fix-$\lambda$, SOCIA-no-CoS).
    → **Revised paper:** `the Ablation findings paragraph in Section 4.4`.

-   **BO vs. SBI as alternative calibration schemas.**
    We state clearly that BO and SBI are alternative (either–or) calibration modes, and explain when each is used.
    → **Revised paper:** `Section 3.3, lines 286-290`.


#### Experiments

-   **Random Search and LR baselines.**
    We clarify that both Random Search and LR are pure parameter-calibration baselines operating on the same fixed simulator structure as SOCIA.
    → **Revised paper:** `Section 4.1, lines 397-402`.

-   **Runtime, simulator calls, and token budgets.**
    We report all the information for transparency.
    → **Revised paper:** `last paragraph in Section 4.1`.

-   **SBI posterior sampling and LLM cache.**
    We standardize on 2,000 posterior samples for SBI, describe the LLM-cache mechanism. In rebuttal, a stability study shows that MAE stabilizes once sample sizes exceed a few hundred.
    → **Revised paper:** `Section 4.1 (posterior sampling paragraph), and Table 1`.

-   **Purpose of Section 4.3 experiments.**
    We add an explicit introductory paragraph explaining that Section 4.3 provides qualitative case studies.
    → **Revised paper:** `Start of Section 4.3`.

-   **Positioning w.r.t. human-designed simulators.**
    We clarify that our experiments focus on automated LLM-based systems while human-designed simulators calibrated with BO/SBI operate in a different regime.


#### Presentation, scope, and notation

-   **SOCIA’s concrete task and output.**
    We clarify early that SOCIA takes a natural-language modeling brief plus observational data as input and outputs an executable, calibrated multi-agent simulator.
    → **Revised paper:** `Introduction (lines 102-108)`.

-   **Scope: ABM focus and possible extensions.**
    We state that the current instantiation targets agent-based and micro-simulation settings, and briefly outline how the same orchestration pattern could extend to future work.
    → **Revised paper:** `lines 129–132 and Appendix K`.

-   **Aggregation operator and objectives.**
    We precisely define the aggregation operator $\mathcal{A}(\cdot)$, the validation objectives $J$, and the metric components.
    → **Revised paper:** `Section 3.1 (lines 203-214)`.

-   **Notation disambiguation for $B$, $\lambda$, $T$, $x$, $y$.**
    We consistently use $B$ for the task blueprint and $\lambda$ for the simulator structure, restrict $T$ to the textual task description (using $N, T_{\text{steps}}$ for sizes/horizons), and define SBI posteriors in terms of observables $y_{\text{obs}}$.
    → **Revised paper:** `Sections 3.1–3.3`.

-   **“Agents” vs. “agents”.**
    We globally distinguish simulation agents (entities in the modeled world) from LLM orchestration agents (DAA, CGA, SEA, FGA) and update the terminology accordingly.
    → **Revised paper:** `Introduction (lines 102-108) and Section 3.1, 3.2`.


----------

We are grateful for the reviewers’ recognition of SOCIA’s ambitious, end-to-end design and its potential impact on simulation-based modeling, and we believe the revised manuscript is both clearer and technically stronger as a result of their feedback.

---

### Meta-Review · Area_Chair_Fvgs · 2026-01-05

**Summary:**

The reviewers recognized SOCIA's ambitious integration of LLM-driven structure synthesis with parameter calibration, with Reviewer 3qgX stating "I really liked this article's full spectrum approach to modelling" and Reviewer t7Ju calling it "a valuable step toward automating simulation-based modeling workflows." However, all reviewers raised significant concerns. Reviewer t7Ju noted "the paper does not yet articulate a clear technical algorithm underlying the structure-parameter co-optimization loop" and that refinement "appears to operate heuristically." Reviewer AoLv found "the introduction is highly generic" with "notation is inconsistent and often undefined." Key issues included: unclear algorithmic details for the outer loop (objective, acceptance, convergence), ambiguous BO/SBI relationship, missing human-designed simulator baselines, inconsistent notation, limited scope clarity (agent-based vs. general simulators), and insufficient computational reporting.

**Reviewer Concerns:**

The authors successfully addressed several technical concerns: the outer loop's objective and greedy acceptance rule are now explicit in Section 3.5 and Algorithm 1, BO and SBI are clarified as alternative modes (Section 3.3, lines 286-290), "agent" terminology is disambiguated, notation is consistent (B for blueprint, λ for structure, y_obs for SBI), and computational costs are reported (40 minutes, ~1M tokens per task). Missing references were added and SBI description corrected to "learn a posterior distribution."

Critical concerns remain unresolved. Reviewer t7Ju's fundamental question about "clear technical algorithm" versus "proof-of-concept system" was not answered: implementation details were provided but theoretical grounding and convergence guarantees beyond budget-based termination are absent. The missing human-designed simulator baseline (Reviewer t7Ju: "omits the central baseline") is explained as "operates in a different regime" but not empirically addressed. The CoS justification relies on observed "rapid saturation" rather than principled analysis. Reviewer AoLv's concerns about unclear task definition were partially addressed through text revisions (lines 103-108), but scope ambiguity persists: extensibility claims to non-agent-based simulators (Appendix K) remain speculative. The SBI sample count explanation (LLM caching) is reasonable but the sensitivity study shows stabilization only at 500-5,000 samples, suggesting the 2,000 choice lacks strong justification.

**Reviewer Scores:**

Reviewer t7Ju (original 2): estimated revised score 4. Technical questions were addressed with concrete details, and the reviewer acknowledged "the idea is promising and potentially impactful." However, the core criticism that "the technical contribution would be significantly strengthened by a more precise definition of the optimization loop" and lack of "what (if anything) can be guaranteed" remains unresolved.

Reviewer AoLv (original 2): estimated revised score 4. Notation inconsistencies were resolved and the introduction now provides task definition (lines 103-108). However, the reviewer's statement "I found it difficult to grasp the task definition and central methodology fully" reflects presentation issues that text revisions cannot fully address, moving the paper from reject to marginally-below-threshold.

Reviewer 3qgX (original 4): estimated revised score 6. The reviewer was already positive, and concerns about Algorithm 1 placement and methodological detail have been addressed through revised Section 3.5. The constructive tone ("Please can you give some more detail") suggests satisfaction with the comprehensive responses.

Average hypothetical score: 4.67. Recommendation: reject.

---

### Decision · Program_Chairs · 2026-01-26

Reject